# Explaining and Mitigating Crosslingual Tokenizer Inequities

**Catherine Arnett**[1], **Tyler A. Chang**[2], **Stella Biderman**[1], and **Benjamin K. Bergen**[2]

[1]EleutherAI, [2]UC San Diego

## Abstract

The number of tokens it takes to encode parallel text in different languages is known to vary. These disparities are called *token premiums*. Having high token premiums leads to less throughput during training and increases costs at inference. In this paper, we show that even after controlling for dataset size, vocabulary size, and data content, monolingual tokenizers exhibit a wide range of token premiums across languages. To understand the cross-linguistic differences that cause these token premiums, we train a suite of approximately 7,000 comparable monolingual tokenizers for 97 languages, manipulating tokenization algorithm, vocabulary size, and dataset size. We measure token premiums and test for a relationship between factors such as data similarity (between tokenizer training and evaluation), vocabulary size, and pre-tokenization. We also investigate the role of language-specific features such as writing system and word length. We find that similarity between training and test data does not impact token premiums, but vocabulary size and pre-tokenization do. While simply increasing vocabulary size does not lead to reduced token premium effects, we can determine an "optimal" vocabulary size for each language to achieve significantly reduced token premium effects. We also train superword tokenizers which allow merges over whitespaces, and we find that they both reduce token premium effects and improve compression overall. Thus, intervening on the vocabulary size or the pre-tokenizer significantly reduces crosslingual token premium effects.

🤗 MonTok Tokenizers  ⭘ Code and Data

## 1   Introduction

Multilingual tokenizers generally require different numbers of tokens to represent parallel text in different languages. In other words, multilingual tokenizers are able to compress text better for some languages than for others. This phenomenon, *token premiums* (Petrov et al., 2023), leads to increased inference costs and latency for languages with high premiums (Ahia et al., 2023; Petrov et al., 2023). In previous work, these effects have been demonstrated only in multilingual tokenizers, which are trained on different proportions of data per language. It is possible, therefore, that these effects could be driven exclusively by the proportion of training data per language. However, in this paper, we show that monolingual tokenizers trained with exactly the same implementation, dataset size, and vocabulary size demonstrate widely variable token premium effects (Fig. 1).

We then ask why tokenizers trained in the same way lead to these different compression rates for different languages. We first determine the effects of byte premiums (variable byte encoding lengths for content-matched text in different languages; Arnett et al., 2024) and tokenizer types (BPE and Unigram) on compression (Section 3). We find that BPE tokenizers have the best compression rates overall, and byte premium scaling of the training dataset size has little effect. We then identify factors that correlate with token premiums (Section 4), including train-eval domain similarity, mean

39th Conference on Neural Information Processing Systems (NeurIPS 2025).

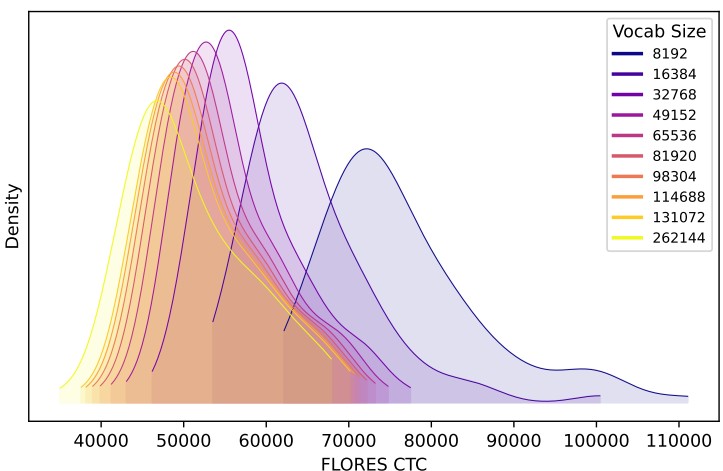

Figure 1: Distribution of corpus token counts (CTCs) for all languages in our sample as a density plot over CTC scores, for the BP-unscaled BPE tokenizers. Higher CTCs indicate worse compression. The distribution for each vocabulary size is plotted separately. As vocabulary size increases, the distribution shifts left and becomes more compressed, and the right tail becomes less extreme.

token length, and proportion of whitespaces in each language. In total, we train approximately 7000 monolingual tokenizers, which we make available on Hugging Face: `https://huggingface.co/datasets/catherinearnett/montok`.

Finally, we explore ways to mitigate token premium inequities by training tokenizers on parallel data, adjusting vocabulary sizes per language, and removing whitespace pre-tokenization (Section 5). We find that training on parallel data does not meaningfully reduce token premium effects. However, we can determine an "optimal" vocabulary size for each language, which significantly reduces token premiums. Additionally, we find that removing whitespace pre-tokenization (*i.e.* SuperBPE tokenizers; Liu et al., 2025) also substantially reduces token premium effects.

In order to fully address token premium effects and design more equitable multilingual tokenizers, we need to understand all the factors that lead to token premium effects. In this work, we show that interactions between tokenizer design and inherent features of a language largely determine differences in compression. These findings represent a key step in understanding the implicit biases that tokenizer design choices might incur.

## 2 Related Work

### 2.1 Token Premiums

Token premiums are the relative differences in the number of tokens used to encode the same content in different languages (Petrov et al., 2023). A higher token premium means that more tokens are needed to encode the same content, which corresponds to lower compression. High token premiums have sometimes been referred to as over-segmentation (Rust et al., 2021) or over-tokenization (Liang et al., 2023). Ahia et al. (2023) and Petrov et al. (2023) showed that multilingual tokenizers often have variable token premiums for the different languages they are trained on.

Token premium effects have only previously been observed in multilingual tokenizers. Multilingual tokenizers are trained on different amounts of data for different languages, and some languages are over- or under-represented in the tokenizer vocabularies (Balhar, 2023). In these tokenizers, therefore, differences in token premiums may be due to training data amounts or vocabulary allocation per language. The current paper, to the best of our knowledge, is the first to systematically document token premiums in comparably trained monolingual tokenizers. In this paper, therefore, we make a distinction between crosslingual and multilingual. By crosslingual, we mean differences between languages that are observed in monolingual settings. We use multilingual to refer to settings where a tokenizer or model is trained on multiple languages.

To reduce token premium inequities, byte-level tokenization (sometimes referred to as "tokenizer-free"; Choe et al., 2019; Clark et al., 2022; Xue et al., 2022, *inter alia*) has been considered as an alternative tokenization approach. However, different languages require different numbers of bytes to encode equivalent content (Arnett et al., 2024), a phenomenon known as "byte premiums". Therefore a byte-level tokenizer, such as that for ByT5 (Xue et al., 2022), still has token premiums—in this case, identical to byte premiums. Some languages have extremely high byte premiums (Burmese: 3.51, Dzhongkha: 3.64; Shan: 3.94; Petrov et al., 2023).

There have been at least two proposed methods to reduce token premiums in multilingual byte-based tokenizers. First, Limisiewicz et al. (2024) proposed MYTE, which replaces long byte strings with unused bytes that represents meaningful word segments, *morphemes*. This method leads to significantly reduced token premium, or byte premium, effects, but it requires a predefined dictionary of morphemes. Relatedly, Ahia et al. (2024) proposed MAGNET, which takes byte-level inputs and uses script-specific boundary-predictor modules to dynamically predict segmentation boundaries. Those boundaries are used to pool together byte-level representations before feeding them into the language model as input. This leads to more equitable segmentation across languages. By contrast, in this paper, we discuss methods to mitigate token premiums in subword tokenizers, without modifying the tokenization algorithm.

## 2.2 Measuring Compression

It is generally desirable for a text to be encoded in as few tokens as possible, *i.e.* to be maximally compressed, and therefore have a low token premium. In this paper, we measure compression using *corpus token count* (CTC; Schmidt et al., 2024), which is the total number of tokens in a corpus. We compute CTC over parallel texts, thereby holding information content constant. As a result, we can compare CTCs across languages as an operationalization of compression, where lower CTC corresponds to higher compression. CTCs can be converted to token premiums as in Petrov et al. (2023) by normalizing them by the CTC of a reference language (e.g. English).

Two other compression metrics are less well suited to the present work. First, fertility, another common metric of compression, takes the average number of tokens per word (Rust et al., 2021). Lower fertility corresponds to better compression. This metric requires a functional definition of wordhood, which does not exist[1]. Even if it did, different languages encode different amounts of information per word. For example, Eastern Canadian Inuktitut has a single word which means "They wanted to catch a walrus" (Gutierrez-Vasques et al., 2023, citing Fortescue, 1992, Dahl, 2017). We thus opt not to use fertility, as it is unclear how to fairly calculate this across different languages. Second, Schmidt et al. (2025) use bytes-per-token as a metric for compression. However, because different languages encode different amounts of information per byte (byte premiums; Arnett et al., 2024), this measure is not suitable for comparing compression across languages.

## 2.3 The Relationship Between Compression and Performance

How compression relates to language model performance on downstream tasks remains uncertain. Higher compression rates mean that sequences of a fixed length contain more information, which may lead to improved model performance (Deletang et al., 2024), and several studies have demonstrated a positive relationship between higher compression and task performance (Goldman et al., 2024; Gallé, 2019). However, through careful manipulation of various aspects of the tokenizer, Schmidt et al. (2024) recently found that there was no relationship between compression and model performance. These experiments are limited to English, and it is thus unclear the extent to which they generalize to other languages. We argue that the statistical properties of tokenizers are important in and of themselves. Higher compression leads to lower training and inference costs due to shorter sequence lengths for the same content. Therefore, compression dictates the computational and fiscal cost of both inference and training over fixed dataset sizes.

---

[1]In English, this is often operationalized as whitespace-separated strings. However, there exist cases, e.g. compound words like 'ice cream', which contain interword whitespaces, but otherwise behave like single words (Zwicky, 1986). Some languages, e.g. Chinese, do not use whitespaces at all. There is not another definition of wordhood that is widely accepted (Haspelmath, 2017).

# 3 Tokenizer Training

For the experiments below, we train a set of subword tokenizers to be comparable across languages, matching training dataset size, vocabulary size, and training method. This allows us to fairly evaluate differences in compression across languages. We use the text datasets from Chang et al. (2024) for all tokenizer training, restricting to the 97 languages with at least 300MB of text data and for which the training data was not significantly contaminated with the FLORES dataset (see Appendix C for details). We use the Hugging Face `tokenizers` (Hugging Face, 2020) package to train these tokenizers. For each language, we train a tokenizer for two tokenizer types (BPE and Unigram; Sennrich et al., 2016; Kudo, 2018) on 300MB of text data, for seven vocabulary sizes ranging from 16384 to 114688. All vocabulary sizes we use are divisible by 128, because this has been shown to lead to better model throughput (Anthony et al., 2024; Groeneveld et al., 2024). We also manipulate whether we scale the tokenizer training dataset size by each language's *byte premium*, the ratio of bytes required to encode content-matched text in each language compared to English. We call this byte-premium scaling (henceforth BP-scaling; Arnett et al., 2024). All 97 languages are listed in Appendix A and additional details about tokenizer training can be found in Appendix B.

One potential concern about our tokenizers is their relatively small training dataset size. Many tokenizers are trained on gigabytes of text, *e.g.* Schmidt et al. (2024). Reddy et al. (2025) recently showed that up to training dataset sizes of 180GB, additional data may lead to better tokenization quality as measured by morphological alignment and correlations with metrics of human language processing. Our tokenizers are trained on substantially less data, in order to prioritize the inclusion of lower-resource languages. 300MB was the largest dataset size available for the language in our sample with the smallest amount of available data. Still, we find that our tokenizers have comparable compression rates to those of OLMo , Pythia, and SmolLM (Groeneveld et al., 2024; Biderman et al., 2023; Allal et al., 2025; see Appendix D). This suggests that our dataset size is not too small to draw conclusions about the role of tokenizer type, vocabulary size, and other tokenizer training decisions on compression.

**Measuring Compression.** We calculate token premiums by calculating CTC on parallel text for the corresponding language for each tokenizer. We use the FLORES-200 dataset (Costa-Jussà et al., 2022), which is a high-quality parallel translation dataset and which includes all 97 languages in our sample. The CTC for each language (for a given tokenizer type, BP-scaling, and vocabulary size) is the un-normalized token premium; our results are equivalent to using the token premiums in Petrov et al. (2023), who normalize by the CTC of a reference language (English). In our results below, a higher CTC indicates *worse* compression.

## 3.1 Effects of Byte Premium Scaling and Tokenizer Type

**Byte Premium Scaling.**  First, we discuss the impact of BP-scaling the tokenizer training dataset size. We fit a linear mixed effects model with the `lme4` package in R (Bates et al., 2014), using CTC as the dependent variable. As predictors, we include a fixed effect of BP-scaling (a binary variable that indicates whether or not the training data is BP-scaled) and random effects for vocabulary size and tokenizer type. We find no effect of BP-scaling the tokenizer training data ($t(3544)$=-0.615, $p$=0.539). We provide further motivation for this analysis and further results in Appendix E. For the remaining experiments, we use tokenizers trained with unscaled training dataset sizes.

**Tokenizer Type.**  Next, we turn to the role of tokenizer type. Comparing our BPE and Unigram tokenizers, we find that across vocabulary sizes, BPE and Unigram show a wide distribution of CTCs and therefore a wide range of token premiums, with BPE exhibiting slightly lower CTCs (*i.e.* better compression) overall. We provide further visualization in Appendix F.

We also compare with the SentencePiece (Kudo and Richardson, 2018) implementation of Unigram tokenizers. We use the tokenizers from the Goldfish models (Chang et al., 2024), which were trained on 5MB of BP-scaled data[2] and with vocabulary size 50000.[3] We evaluate all the Goldfish tokenizers for which there exists a corresponding FLORES dataset, for a total of 183 tokenizers.

---

[2]Though we noted that we are using non-BP-scaled training data throughout, because we found no effect of BP-scaling on compression, we do not expect this to impact the results.

[3]We note that our BPE and Unigram tokenizers have vocabulary sizes of 49152, while the SentencePiece tokenizers have vocabulary sizes of exactly 50000.

Figure 3 (left) shows the distribution of CTCs for the tokenizers of each type. All tokenizer types show a wide spread of CTCs across languages, though the absolute CTCs vary. BPE tokenizers demonstrate the most compact distribution and the lowest overall CTCs. This is consistent with previous results in English contexts showing that BPE offers more compression than other tokenizers (Gallé, 2019; Schmidt et al., 2024), but to the best of our knowledge this has not been shown to be true crosslinguistically. SentencePiece implementation of Unigram has the worst compression rates of all tokenizer types tested.

## 3.2 Final Tokenizer Setup

The results in this section suggest that the wide range in token premiums across languages is not unique to any tokenization algorithm, but BPE does exhibit slightly reduced cross-linguistic differences in CTC as well as lower CTCs (better compression) overall. Therefore, for the remaining analyses, we use only the BPE tokenizers with BP-unscaled training dataset sizes. We train tokenizers with an even wider range of vocabulary sizes (from 8192 to 262144), which we use in the experiments in Section 4.

## 4 Explaining Token Premiums

In the previous section, we showed that tokenizers trained on the same amount of data, with the same vocabulary size, and with the same training procedure, there is still show a wide range of compression rates across languages. In this section, we test various potential factors which may drive these crosslinguistic differences in token premiums. We fit linear regressions for each of the factors and report the amount of variance explained by the top predictors (Table 1).

| Predictor | $R^2$ |
|---|---|
| Data Similarity | 0.239 |
| Mean Token Length (vocab) | 0.168 |
| Proportion Whitespaces | 0.157 |
| **Combined** | **0.297** |

Table 1: $R^2$ values for linear regressions predicting FLORES CTC from different predictors, for tokenizers with a vocabulary size of 65536. Values for all predictors are provided in Appendix H.3.

**Data Similarity.**   The datasets we trained our tokenizers on come primarily from web text (Chang et al., 2024), but the domain and quality may vary widely across languages, especially for lower-resource languages (Kreutzer et al., 2022). If the training data is low-quality or contaminated with data from other languages, this may affect the ability of the tokenizer to learn a vocabulary that leads to effective compression. Additionally, if the training data differs too much from the evaluation dataset, e.g. as measured by word overlap, the tokenizer may also learn a sub-optimal vocabulary (Dagan et al., 2024). Here, we measure train-eval data similarity as the number of overlapping tokens between the training and evaluation datasets, normalized by the vocabulary size. Data similarity is the factor that explains the most variance in CTCs, suggesting that train-eval data similarity may be important for token premiums. In Section 5.1, we train tokenizers on parallel data and test whether this reduces token premiums.

**Mean Token Length.**   Mean token length, which we measure in in characters, is directly related to compression, as in order to represent a text with fewer tokens each token must be longer. We calculate mean token length in two different ways. First, we calculate the mean length of each token in the tokenizer's vocabulary (this is the metric reported in Table 1). Second, we calculate the mean token length when only including the tokens used to tokenize the FLORES dataset for each language. The latter should be closely linked with FLORES CTC, as it measures token lengths in the evaluation corpus itself.

We find that mean token length over the vocabulary is not a significant predictor of CTC, but mean token length over FLORES has an $R^2$ of 0.168. This suggests that differences in CTC are not due to

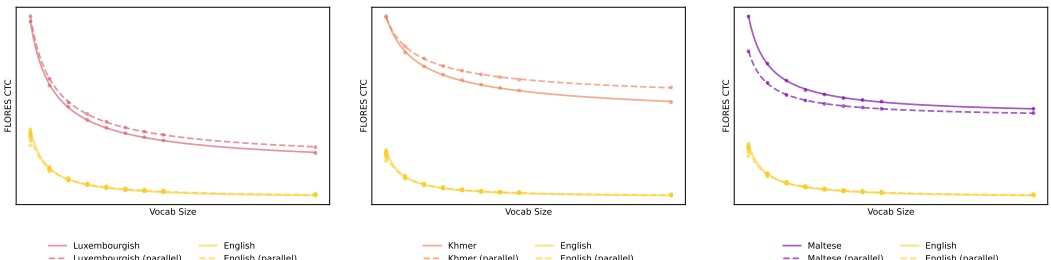

Figure 2: Comparison of CTCs for tokenizers trained on non-parallel (solid lines) versus parallel (dashed lines) data for three languages: Luxembourgish, Khmer, Maltese. Figures for the remaining languages are in Appendix I.

shorter or longer tokens overall in the tokenizer. Instead, differences in CTC are related specifically to the length of tokens that are actually used in the evaluation dataset; tokenizers with high CTCs may have longer vocabulary items in their vocabularies, but they are not being used as frequently. To mitigate CTC differences due to token length, we hypothesize that increasing the tokenizer vocabulary size should increase mean token length and therefore lower CTC. In Section 5.2, we test not only the effect of increasing vocabulary size, but also of identifying an "optimal" vocabulary size for each language, on token premium effects.

**Proportion of Whitespaces.** Most BPE tokenizers use a pre-tokenization step, which splits text into pre-tokens. During training, the tokenizer cannot learn merges that cross the pre-token boundaries. One of the most common pre-tokenization steps splits texts at every whitespace. Languages differ in how many whitespaces (equivalently, how many space-separated "words") they use to encode the same information (see examples in Appendix G). This means that whitespace pre-tokenization could affect different languages to different extents. Indeed, when we measure the proportion of whitespaces in FLORES for each language, we find that it explains a substantial amount of variance in CTCs (Table 1). Motivated by this finding, in Section 5.3, we train "superword" tokenizers (Liu et al., 2025; Schmidt et al., 2025) that allow merges across whitespaces.

**Other Metrics.** We also test various other metrics related to a language and its script (writing system), including number of unique phonemes, entropies over characters and bigrams (character pairs), and a language's script itself when predicting CTCs. We provide more detailed discussion and report full results for all metrics in Appendix H. In the current section, we focused on data similarity, mean token length, and proportion of whitespaces, because these metrics are predict significant amounts of variance. Next, we propose interventions for each of these factors.

## 5 Mitigating Token Premiums

### 5.1 Data Quality and Data Similarity

If either data quality or train-eval data similarity is driving the token premiums, then we should see that differences in CTC decrease when tokenizers are trained on parallel text. Parallel text is content-matched and therefore has identical domain overlap with the evaluation dataset. To test this, we selected seven languages[4] which had the highest CTCs at at least one vocabulary size. We did this in order to select the languages for which there were the greatest token premium effects, to provide more opportunity for a reduction of the effects. We take the parallel data for each target language and English in NLLB (Costa-Jussà et al., 2022), and we create 300MB subsets for each language. Because NLLB is not multi-parallel, we have a different English dataset for each target language. We train new BPE tokenizers for each new parallel dataset using the same vocabulary sizes as described in Section 3.2. We calculate CTC in the same way as previous sections, and we compare the CTCs from the original monolingual tokenizers with those from the tokenizers trained on parallel data. We show the results for three languages in Figure 2.

---

[4]These languages are Cebuano, Khmer, Luxembourgish, Mandarin Chinese, Maltese, Japanese, and Burmese.

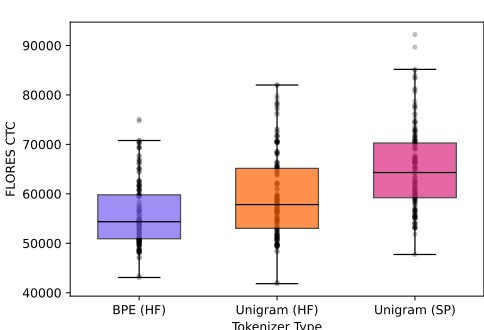
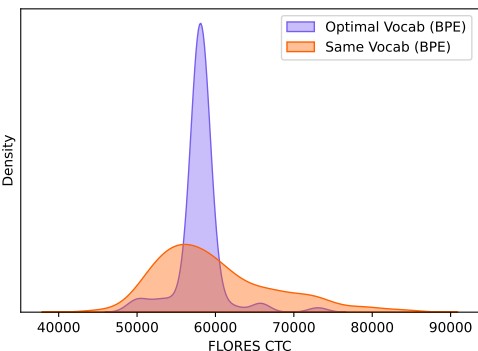

Figure 3: **Left**: the distribution of CTCs for BPE and Unigram tokenizers using Hugging Face (HF) or SentencePiece (SP) implementations, all with vocabulary sizes of approximately 50000. **Right**: the distribution of CTCs for same-vocabulary and optimal-vocabulary BPE tokenizers. The optimal-vocabulary tokenizers have target CTCs of 56000. The same-vocabulary tokenizers have a vocabulary size of 32768.

Comparing the original and parallel-data tokenizers for the seveb languages in our sample, we find that the parallel-data tokenizers have a statistically significant reduction in token premium effects (paired $t$-test; $t(152)=-2.356$, $p=0.0197$). The difference between the CTCs is quite small, however: on average about 1% of the total CTC. We also find that the majority of this effect is driven by the tokenizers with the smallest vocabulary size (16384 for these tests). Statistical tests per vocabulary size are reported in Appendix I.2. Overall, these results indicate that training on parallel data has a small effect on CTC, but it does not meaningfully reduce token premium effects.[5]

## 5.2 Vocabulary Size

Next, we evaluate whether different vocabulary sizes have greater or lesser token premium effects. While increasing vocabulary size reduces CTC overall, even the largest vocabulary size has large token premium effects (Figure 1). We test the difference in variance between CTC distributions for the smallest and largest vocabulary sizes with a Fisher-Snedecor test (F-test), and we find no significant difference ($F_{96,96}=1.125$, $p=0.565$).

Then, we test whether we can find an "optimal" vocabulary size per language such that there is a reduction in variation of CTCs. To do this, we fit a power law curve for the relationship between CTC and vocabulary size for each language (e.g. the curves in Figure 2). Using these curves, for each language, we predict the vocabulary size at which the tokenizer will reach a given CTC. We then train new tokenizers at the "optimal" vocabulary size for a given target CTC. We provide more detail on the training of these tokenizers and validation of our estimation method in Appendix J.

Tokenizers with optimal vocabulary sizes per language have reduced token premium effects compared to tokenizers with the same vocabulary size across languages (Figure 3; right). We find the optimal-vocabulary tokenizers to have significantly less variance in CTCs (F-test; $F_{80,387}=0.150$, $p<0.001$). For BPE tokenizers, therefore, different languages need to be allocated different vocabulary sizes to achieve more comparable compression.

## 5.3 Whitespace Pre-Tokenization

Finally, we intervene on the effect of whitespaces. To do this, we train superword tokenizers, which allow merges over whitespaces. Therefore, tokens may be composed of multiple whitespace-separated words and the effect of whitespace pretokenization on compression should be reduced. We use the SuperBPE (Liu et al., 2025) implementation. Our tokenizers are the first non-English superword tokenizers to the best of our knowledge. We train only tokenizers for vocabulary sizes over 64000, as

---

[5]We test whether the parallel-data tokenizers have higher data similarity between training and evaluation data (as measured in Section 4). They do (paired $t$-test; $t(270.55)=1.564$, $p=0.119$). This may explain why this intervention did not lead to greatly reduced token premium effects.

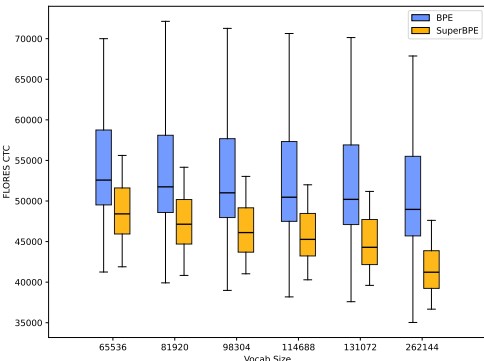
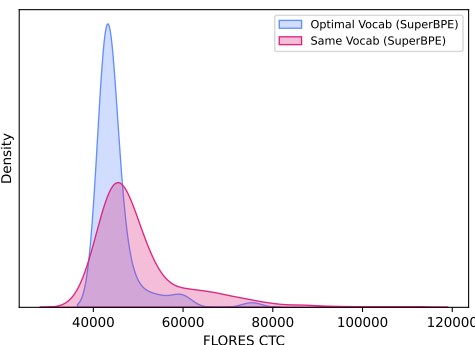

Figure 4: **Left**: the distribution of CTCs for BPE and SuperBPE tokenizers for each vocabulary size. **Right**: the distribution of CTCs for same-vocabulary and predicted CTCs for the optimal-vocabulary SuperBPE tokenizers. The optimal-vocabulary tokenizers have target CTCs of 43000. The same-vocabulary tokenizers have a vocabulary size of 49152.

we predict the superword tokenizers need larger vocabulary sizes to learn longer superword tokens. Thus, we expect to see the biggest differences between BPE and SuperBPE tokenizers at the higher vocabulary sizes. We include additional details about training SuperBPE tokenizers in Appendix K.

We find that SuperBPE tokenizers demonstrate lower average CTCs and less variance in CTCs at every vocabulary size (Table 10 in Appendix L). We plot the distributions in Figure 4. We also run a linear regression to test the relationship between proportion of whitespaces and CTC (as in Section 4), but find that there is still a significant relationship at all vocabulary sizes except 65536 (full results in Table 11 in Appendix L).

We then test whether setting optimal vocabulary sizes for each language would lead to reduced token premium effects in SuperBPE tokenizers. We use the same method to determine the optimal vocabulary size for each language as we did for BPE (see Appendix M). We did not train SuperBPE tokenizers for each optimal vocabulary, however we demonstrated that our CTC estimation method was accurate for BPE (Appendix J.3). We estimate there to be significantly less variance in CTC for the optimal-vocabulary set than the same-vocabulary set for SuperBPE ($F$-test; $F_{558,76}$=3.8727, $p$<0.001). This result suggests that training optimal vocabulary tokenizers for SuperBPE would lead to further reduced token premium effects (Figure 4; right).

### 5.4 Remaining Variance

There are still significant effects of length ratio and bytes-per-character after the interventions on whitespace pre-tokenizers and optimal vocabulary sizes (results in Appendix N). These are features which are determined by inherent properties of the languages, the writing system, the way the writing system is encoded in UTF-8, and their interactions. Future work can seek ways to address remaining inequities, especially those introduced by UTF-8. Recent work has introduced a novel encoding scheme to replace UTF-8 in the context of tokenization, in which all Unicode characters are represented with the same number of byte-equivalent units (Land and Arnett, 2025). This encoding scheme does not address any differences introduced by the length ratios, however.

## 6 Discussion

**The Impact of Pre-Tokenization.** Whitespace pre-tokenization is responsible for a large portion of the crosslinguistic differences in compression. This is consistent with recent work that showed that pre-tokenization had a larger impact on model performance than any other tokenizer design choice tested, including the training corpus and vocabulary size (Wegmann et al., 2025). Relatedly, Velayuthan and Sarveswaran (2025) show that many common pre-tokenizers segment text below the grapheme level for multiple South Asian languages (Hindi, Tamil, Sinhala). In this case, this is due to regular expression pre-tokenizers, which often pre-tokenize on punctuation and other symbols. This may include common diacritics and sub-graphemic units in non-Latin scripts, leading to catastrophic

compression for some languages. The work in this paper contributes to a growing body of work that shows the unequal impact of pre-tokenizer design decisions crosslinguistically.

**Language-Specific Considerations for Language Model Design.** We show that there is no vocabulary size at which monolingual tokenizers will achieve similar compression across a wide variety of languages. However, we can overcome this through setting language-specific vocabulary sizes. Vocabulary size, therefore, should perhaps be a design choice that varies according to the target language or languages. Relatedly, Chang et al. (2024) argue that it may be necessary to scale language model training data in order to account for byte premiums and achieve comparable performance across languages (Chang et al., 2024). The same settings that work well for English will not necessarily work equally well for all languages.

**Too Much Compression?** One concern is that some of the tokenizers we train—especially the SuperBPE tokenizers—provide *too much* compression. While there has been work discussing the relative benefits (Gallé, 2019) (or lack thereof; Schmidt et al., 2024) of increased compression, there has been less work on the negative impacts of increased compression. Perhaps too much compression could be giving the model effectively less time to think, by dedicating fewer FLOPS to predicting each next token. Goyal et al. (2024) found that adding "pause" tokens allowed models to perform additional computations before providing an answer and found that this increased performance.

**Applications to Multilingual Tokenization.** These findings inform future development of more equitable multilingual tokenizers. As it is increasingly common to train language models on multilingual data, it is more important than ever to reconsider decisions that might be disadvantaging performance or increasing cost for some languages more than others. Simply allocating more vocabulary to some languages may drastically improve compression. Vocabulary allocation, or how much of the vocabulary of a multilingual tokenizer is dedicated to a particular language, is correlated with model performance (Limisiewicz et al., 2023). Our results suggest that some languages might benefit from an increase in vocabulary allocation more than others.

Petrov et al. (2023) proposed training monolingual tokenizers with similar compression and combining them to achieve more balanced token premiums. Our work shows that if you did this with monolingual tokenizers with the same vocabulary size, this would likely not mitigate token premium effects. However, if this method were employed with optimal-vocabulary-size tokenizers for each language, it might lead to reduced token premium effects.

**The Link Between Compression and Performance.** We have thoroughly tested the relationship between different language features and compression, but we have not directly linked this to model performance. Token premiums affect the cost of training and inference, as they determine the number of computations needed for a text of a fixed length. Therefore, we believe the results of this paper are informative, even if they do not impact model performance. While for English, small changes in compression may not be correlated with better performance (Schmidt et al., 2024), monolingual English tokenizers have among the lowest CTCs (Figure 13). Thus, differences in compression may be smaller relative to other languages. For languages with higher CTCs, improvements in compression may be more drastic and may lead to significant performance gains.

Additionally, there is likely to be a complex relationship between compression and other tokenizer properties, which may impact performance. We showed that Unigram tokenizers showed the worst compression across languages, but Bostrom and Durrett (2020) showed that Unigram tokenizers segment text into more linguistically-aligned units than BPE. It remains unclear, however, whether having linguistically-aligned tokens is clearly linked to better language model performance (Arnett and Bergen, 2025).

# 7 Conclusion

This work investigates how different language-specific features interact with common tokenization algorithms. First, we show that training monolingual tokenizers in the same way leads to different compression rates for different languages, *i.e.* token premium effects. We identify three main factors which might lead to this effect. First, we ask whether these effects are due to differences in the training data. After training tokenizers on parallel data, we find that data similarity does not meaningfully

explain crosslingual differences in CTC. Second, we test whether manipulating vocabulary size decreases differences in compression. Increasing vocabulary size does not decrease token premium effects; however, we show that training a tokenizer with an "optimal" vocabulary size for each language drastically reduces differences in CTC. Finally, we analyze the effect of pre-tokenization. Different languages use varying amounts of whitespaces, meaning that whitespace pre-tokenization affects languages differently. We train superword tokenizers, which allow merges over whitespace boundaries. This not only reduces token premium effects, but overall improves compression. Together, these results help to inform the development of efficient and equitable tokenizers for more languages.

## 8    Limitations

While we aimed to be comprehensive in our experiments and to carefully manipulate all variables, some of our experiments were limited in breadth. For the tokenizers we trained on parallel data, we could not train on as many languages due to the availability of parallel data. Additionally, for each language, we had to train a new baseline tokenizer, doubling the compute and data requirements for that experiment. For the parallel-data tokenizers, we were also not able to compare across languages, but only between a target language and an English baseline. This was due to the availability of parallel data, which is even more limited than multilingual pre-training data. This could potentially also be a limitation of the analysis. It is unclear whether comparisons with different languages would lead to different results.

While we showed that we achieved similar compression to pre-trained tokenizers with our small dataset size, it may be the case that the dataset sizes we used were too small for the SuperBPE tokenizers. It could be the case that SuperBPE tokenizers need more data than BPE tokenizers, in order to prevent overfitting to the training data, especially for longer and lower-frequency tokens.

We solely used FLORES as the evaluation dataset for all of our experiments. FLORES is a dataset created by translating from English into each of the languages represented in the dataset. Therefore translation artifacts or differences in translation quality may affect our results.

Relatedly, we measured data similarity as the number of overlapping tokens, which could potentially be measuring token diversity in addition to or instead of data similarity. This could explain why we did not see large changes in data similarity between the non-parallel-data and parallel-data tokenizers.

Finally, we did not train any language models with our tokenizers, therefore we cannot make any claims about the connection between our findings and changes in language model performance.

## Acknowledgments

We would like to thank the UC San Diego Social Sciences Computing Facility Team for the use of the Social Sciences Research and Development Environment (SSRDE) cluster. Tokenizers were trained and evaluated using hardware provided by the NVIDIA Corporation as part of an NVIDIA Academic Hardware Grant. We thank Alisa Liu and Jonathan Hayase for their assistance with the crosslingual implementation of SuperBPE.

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

# A   Language Sample

Table 2 lists the 97 languages in our sample, along with their language family and ISO codes (comprised of the ISO 639-3, which denotes language, and ISO 15924 codes, which denotes script).

Table 2: Languages with their Codes and Families

| Language Code | Language Name | Language Family |
| --- | --- | --- |
| afr_latn | Afrikaans | Indo-European |
| als_latn | Tosk Albanian | Indo-European |
| amh_ethi | Amharic | Afro-Asiatic |
| arb_arab | Modern Standard Arabic | Afro-Asiatic |
| asm_beng | Assamese | Indo-European |
| bak_cyrl | Bashkir | Turkic |
| bel_cyrl | Belarusian | Indo-European |
| ben_beng | Bengali | Indo-European |
| bos_latn | Bosnian | Indo-European |
| bul_cyrl | Bulgarian | Indo-European |
| cat_latn | Catalan | Indo-European |
| ceb_latn | Cebuano | Austronesian |
| ces_latn | Czech | Indo-European |
| ckb_arab | Central Kurdish | Indo-European |
| cym_latn | Welsh | Indo-European |
| dan_latn | Danish | Indo-European |
| deu_latn | German | Indo-European |
| ell_grek | Greek | Indo-European |
| eng_latn | English | Indo-European |
| epo_latn | Esperanto | Constructed |
| est_latn | Estonian | Uralic |
| eus_latn | Basque | Language Isolate |
| fao_latn | Faroese | Indo-European |
| fin_latn | Finnish | Uralic |
| fra_latn | French | Indo-European |
| gla_latn | Scottish Gaelic | Indo-European |
| gle_latn | Irish | Indo-European |
| glg_latn | Galician | Indo-European |
| guj_gujr | Gujarati | Indo-European |
| hat_latn | Haitian Creole | Creole |
| hau_latn | Hausa | Afro-Asiatic |
| heb_hebr | Hebrew | Afro-Asiatic |
| hin_deva | Hindi | Indo-European |
| hrv_latn | Croatian | Indo-European |
| hun_latn | Hungarian | Uralic |
| hye_armn | Armenian | Indo-European |
| ibo_latn | Igbo | Niger-Congo |
| ind_latn | Indonesian | Austronesian |
| isl_latn | Icelandic | Indo-European |
| ita_latn | Italian | Indo-European |
| jav_latn | Javanese | Austronesian |
| jpn_jpan | Japanese | Japonic |
| kan_knda | Kannada | Dravidian |
| kat_geor | Georgian | Kartvelian |
| kaz_cyrl | Kazakh | Turkic |
| khm_khmr | Khmer | Austroasiatic |
| kir_cyrl | Kyrgyz | Turkic |
| kin_latn | Kinyarwanda | Niger-Congo |
| kor_hang | Korean | Koreanic |

Table 2: Languages with their Codes and Families (continued)

| Language Code | Language Name | Language Family |
|---|---|---|
| lao_laoo | Lao | Tai-Kadai |
| lit_latn | Lithuanian | Indo-European |
| ltz_latn | Luxembourgish | Indo-European |
| mal_mlym | Malayalam | Dravidian |
| mar_deva | Marathi | Indo-European |
| mkd_cyrl | Macedonian | Indo-European |
| mlt_latn | Maltese | Afro-Asiatic |
| mri_latn | Māori | Austronesian |
| mya_mymr | Burmese | Sino-Tibetan |
| nld_latn | Dutch | Indo-European |
| nno_latn | Norwegian Nynorsk | Indo-European |
| nob_latn | Norwegian Bokmål | Indo-European |
| nya_latn | Nyanja(Chewa, Chichewa) | Niger-Congo |
| oci_latn | Occitan | Indo-European |
| pan_guru | Punjabi | Indo-European |
| pes_arab | Persian | Indo-European |
| plt_latn | Plateau Malagasy | Austronesian |
| pol_latn | Polish | Indo-European |
| por_latn | Portuguese | Indo-European |
| ron_latn | Romanian | Indo-European |
| rus_cyrl | Russian | Indo-European |
| sin_sinh | Sinhala | Indo-European |
| slk_latn | Slovak | Indo-European |
| slv_latn | Slovenian | Indo-European |
| sna_latn | Shona | Niger-Congo |
| snd_arab | Sindhi | Indo-European |
| som_latn | Somali | Afro-Asiatic |
| sot_latn | Sotho | Niger-Congo |
| spa_latn | Spanish | Indo-European |
| srp_cyrl | Serbian | Indo-European |
| sun_latn | Sundanese | Austronesian |
| swe_latn | Swedish | Indo-European |
| tam_taml | Tamil | Dravidian |
| tat_cyrl | Tatar | Turkic |
| tel_telu | Telugu | Dravidian |
| tgk_cyrl | Tajik | Indo-European |
| tgl_latn | Tagalog (Filipino) | Austronesian |
| tha_thai | Thai | Tai-Kadai |
| tur_latn | Turkish | Turkic |
| uig_arab | Uyghur | Turkic |
| ukr_cyrl | Ukrainian | Indo-European |
| urd_arab | Urdu | Indo-European |
| vie_latn | Vietnamese | Austroasiatic |
| xho_latn | Xhosa | Niger-Congo |
| yor_latn | Yoruba | Niger-Congo |
| zho_hans | Chinese (Simplified) | Sino-Tibetan |
| zsm_latn | Malay | Austronesian |
| zul_latn | Zulu | Niger-Congo |

# B  Tokenizer Training Details

**Tokenizer Data and Language Sample.** We create training datasets of 300MB, sourced from Chang et al. (2024). Links to individual source corpora are included at `https://github.com/tylerachang/curse-of-multilinguality`. We do not redistributed the compiled datasets, as the original source dataset licenses do not permit us to do so. However, all datasets are publicly available.

For each language, we also create a dataset where the dataset size is scaled according to its byte premium (Arnett et al., 2024). Byte-premium scaling (BP-scaling) the data is intended to achieve more comparable information content in datasets across languages. There are 98 languages with at least 300MB of BP-scaled data from Chang et al. (2024). However, we exclude Samoan, because we found the training data to be severely contaminated with portions of FLORES (Appendix C). We list all 97 languages included in our experiments in Appendix A. For languages with byte premiums over 1 and therefore a BP-scaled dataset of over 300MB, the unscaled dataset is a subset of the scaled dataset. For languages with a byte premium less than one, this is not the case, as the unscaled dataset is larger than the scaled dataset. While the datasets are not parallel in content, the datasets are generally comprised of text from similar genres and domains.

All tokenizers were trained using the CPUs from one server equipped with an NVIDIA RTX A6000.

**Tokenizers.** For each language and dataset size, we train tokenizers which differ in vocabulary size and tokenizer type. We use seven vocabulary sizes: 16384, 32768, 49152, 65536, 81920, 98304, 114688. These vocabulary sizes were chosen to cover the range of frequently used vocabulary sizes (between 32000 and 64000) as well as additional vocabulary sizes above and below this range. The vocabulary sizes constitute an arithmetic progression, where each vocabulary size is 16384 higher than the previous one, with the exception of the largest size, which is double the next smallest size. The smallest vocabulary size we use is 16384, because the smaller vocabulary size we tried, 8192, was too small forthe Unigram tokenizers to converge in many languages. All vocabulary sizes are divisible by 128, following Anthony et al. (2024) and Groeneveld et al. (2024), which makes our tokenizers more suitable for training language models in the future. For each vocabulary size, we train both BPE (Gage, 1994; Sennrich et al., 2016) and Unigram (Kudo, 2018) tokenizers. For each language (n=98[6]), we train a tokenizer for each possible combination of vocabulary size (7), tokenizer type (2), and BP-scaling (2) for a total of 2,744 tokenizers. We release all tokenizers and training datasets Hugging Face[7]. Code for training and evaluation is released: `https://github.com/catherinearnett/explaining_tokenizer_inequities`.

# C  Contamination Analysis

Our training data are subsets of the training data from Chang et al. (2024), who claim to exclude the FLORES dataset from their training datasets, but there is still the possibility that FLORES could be contaminated inadvertently in the web data sources. To check for any potential contamination of FLORES in the datasets, we tokenize each FLORES sequence and the entire training dataset from Chang et al. (2024) for each language that is in both their dataset and FLORES (208 languages, which is a superset of the languages used in the analyses in this paper). We compute the total number of times that the first 10 tokens of any FLORES example appears in the training dataset for the language. For 72% of languages, no FLORES examples appear in the training dataset at all. For 98% of languages, less than 10 FLORES examples appear in the training dataset (out of over 2000 FLORES examples total). The only language with notable FLORES contamination is `smo_latn` (Samoan; 7155 occurrences of FLORES examples in the training dataset). As it is an extreme outlier in the contamination analyses, we do not include it in the analyses in this paper.

# D  Comparing CTC to Pre-trained Tokenizers

In order to determine that our training data size is not too small, we compare the FLORES CTC of existing English-centric tokenizers, OLMo (Groeneveld et al., 2024), Pythia (Biderman et al., 2023),

---

[6]We then exclude Samoan per Appendix C, for a total of 97 languages included in our analyses.

[7]`https://huggingface.co/datasets/catherinearnett/montok`

and SmolLM (Allal et al., 2025), all of which have vocabulary sizes of approximately 50000[8]. These are represented by dashed horizontal lines (Figure 5). We compare these to the BPE and Unigram tokenizers we trained on English data (solid lines; Fig. 5). We note that our BPE tokenizers in particular have lower token premiums than all three tokenizers at most vocabulary sizes.

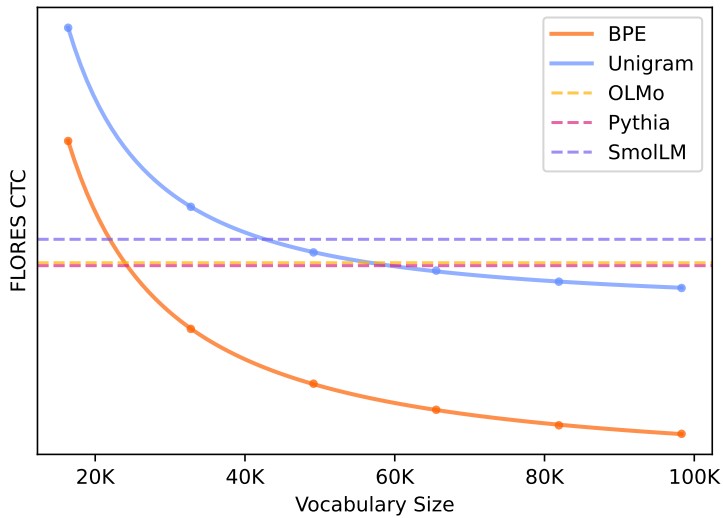

Figure 5: The solid curves, fit using a power law, represent the measured CTCs for each vocabulary size, plotted as points along the curve. The horizontal dashed lines indicate the CTCs of the OLMo, Pythia, and SmolLM tokenizers.

## E  BPE vs. Unigram Tokenizers

We ask whether there is a direct relationship between byte and token premiums. We hypothesize that if there is, it would only be for Byte-pair Encoding (BPE) tokenizers. Byte premiums capture the observation that some languages need more bytes to convey the same amount of information compared to other languages. BPE tokenizers work by first splitting a sequence into individual bytes. The tokenizer learns to merge pairs of bytes iteratively. For a given token, the more bytes it is composed of, the more merges the tokenizer has to execute in order to create that token. For languages with higher byte premiums, for a sequence with a fixed amount of information, a BPE tokenizer will have to execute more merges in order to create tokens with the same amount of content. Therefore, for languages with higher byte premiums, BPE tokenizers will be less likely to learn the correct number of merges, and thus will exhibit stronger token premium effects across monolingual tokenizers trained on the same amount of data for each language.

Unigram, by contrast, during training creates the vocabulary by first adding each unique whitespace-separated word to the vocabulary. It then iteratively removes items until the vocabulary is the desired size. The final vocabulary maximizes the unigram token probability over the training corpus. Therefore, we hypothesize that byte premiums should not directly impact Unigram tokenization.

In Section 3.1, we tested whether byte premium scaling the training data affects CTC. Here, we test the hypothesis about the effect of byte premiums on CTC and its interaction with tokenizer type.

**Do tokenizers show a relationship between byte premium and token premium?**  We fit a linear mixed effects model, predicting token premium with a fixed effect of byte premium and random intercepts for both vocabulary size and tokenizer type. We test this only for tokenizers trained on unscaled data. We find an extremely small, but significant, *negative* relationship between byte premium and token premium ($\beta$ = -0.0418, SE = 0.00542, $t(1362)$ = -7.712, $p < 0.001$), such that as byte premium goes up, token premiums go down. This is the opposite of what was predicted by our hypothesis. The effect can be interpreted such that as byte premium goes up by one (which is a

---

[8]OLMo's vocabuary size is 50279, Pythia's is 50253, and SmolLM's is 49152.

relatively large difference, equivalent to different in byte premium between English and Greek), the CTC goes down by 3.7%. Due to this extremely small effect size, this result does not support our hypothesis that there is a correlation between byte premium.

**Is there a difference in the relationship between byte and token premiums between BPE and Unigram tokenizers?** Our hypothesis predicts that there is a stronger (positive) correlation between byte and token premiums for BPE tokenizers than for Unigram tokenizers. We fit a linear mixed effects model predicting token premiums with both byte premium and tokenizer type as fixed effects. We also fit this model so that we test for the interaction between byte premium and tokenizer type. As in the previous linear mixed effects model, we fit a random intercept for vocabulary size. While we find the effect of byte premium to be significant ($\beta$=-0.0350, SE=0.00676, $t$(1553)=-5.166, $p$<0.001), there is no significant interaction between byte premium and tokenizer type ($\beta$=-0.0156, SE=0.0102, $t$(1553)=-1.528, $p$=0.127). We also note that there is no significant difference between token premiums between the BPE and Unigram tokenizers ($t$-test; $t$(1553)=-1.528, $p$=0.127). Therefore, it does not seem to be the case that BPE is more sensitive to byte premium effects in terms of token premiums.

# F  Comparison of Tokenizers by Vocabulary Size

We compare the Unigram and BPE tokenizers across all vocabulary sizes (Figure 6). Unigram shows a similar pattern, where increased vocabulary size does not lead to decreased token premiums across languages.

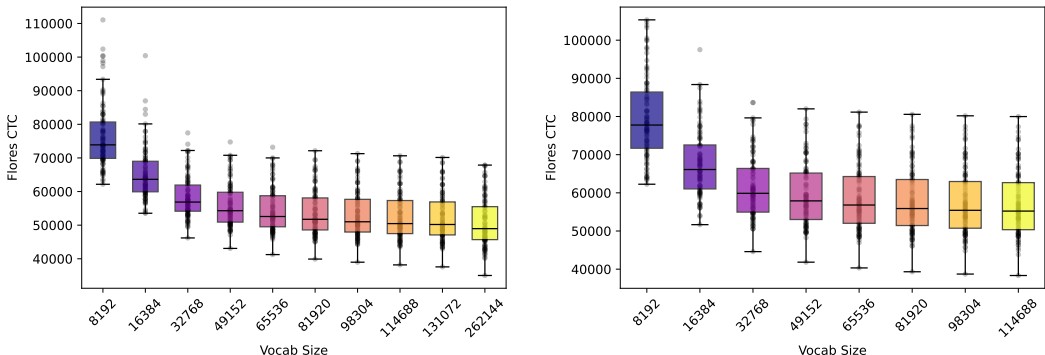

Figure 6: Left: BPE CTC distribution by vocabulary size. Right: Unigram CTC distribution by vocabulary size.

# G  Example FLORES Texts

Below, we include examples from the FLORES dataset for three languages: Scottish Gaelic, Finnish, and English. Each of these sentences are translation equivalents of each other. We observe that languages like Gaelic have higher occurrences of very short words (1-3 characters) and more whitespaces.

# H  Analyses of Additional Metrics

In Section 4, we test a variety of language features. Here, we explain what each of the metrics indexes and how the metrics were calculated (Appendices H.1 and H.2). We also report the correlation metrics for each vocabulary size. We report full results for each metric at each vocabulary size in Appendix H.3.

## H.1  Factors Impacting Sequence and Word Length

We hypothesize that word length should be correlated with compression for a related reason to proportion of whitespaces. Languages with longer words are likely to have fewer words, and—for

| Language | Sentence |
|---|---|
| **Scottish Gaelic** | Dh'innis luchd-saidheans o Sgoil an Leigheis aig Oilthigh Stanford Diluain gun do dh'innlich iad inneal diagnosachd ùr a tha comasach air ceallan a sheòrsachadh: sgealb chlò-bhuailte bheag bhìodach as urrainnear clò-bhualadh air clò-bhualadair ince àbhaisteach 's a chosgas mu aon seant Aimeireaganach gach tè. |
| **Finnish** | Stanfordin yliopiston lääketieteen laitoksen tutkijat ilmoittivat maanantaina uuden diagnostiikkatyökalun keksimisestä: solut tyypin mukaan lajitteleva pienenpieni tulostettava siru, joka voidaan valmistaa normaaleilla mustesuihkutulostimilla mahdollisesti noin yhden Yhdysvaltain sentin kappalehintaan. |
| **English** | On Monday, scientists from the Stanford University School of Medicine announced the invention of a new diagnostic tool that can sort cells by type: a tiny printable chip that can be manufactured using standard inkjet printers for possibly about one U.S. cent each. |

Table 3: Example from FLORES for three languages: Scottish Gaelic, Finnish, and English.

languages that use whitespaces—fewer whitespace-separated words. This, therefore would also be impacted by whitespace pre-tokenization. Therefore, we expect that languages with longer words will have better compression.

**Number of Phonemes.** As not all languages use whitespaces to indicate word boundaries and the notion of wordhood is a fraught concept in linguistics[9], we calculated four metrics, which we believe to be related to word length. The first relates to differences in sound systems between languages. There has been an observed relationship between the number of unique sounds (called *phoneme inventory*) and the average word length across languages. Nettle (1995) found that in a small sample of languages, phoneme inventory was predicted to correlate with word length. More recently, Fenk-Oczlon and Pilz (2021) found that phoneme inventory size was negatively correlated with word length as measured by number of syllables, but not by number of phonemes.

We calculated phoneme inventory size for each language in our sample using PHOIBLE (Moran et al., 2014), a phonological inventory database. We predicted that there would be an inverse relationship between phoneme inventory and word length, and therefore larger inventory size should correspond to lower CTC.

**Number of Characters and Character Distribution.** Following this logic, we also hypothesized that languages with smaller inventories of characters would have longer words and sequences, because you need longer sequences to get enough unique sequences to represent all the words in a language. The benefit of using this metric over phoneme inventory size is that it does not assume that longer words in number of sounds will correspond directly to sequence length in characters, as writing systems differ wildly in the information they encode. Different writing systems may encode each sound roughly (alphabets), entire syllables (abugidas), just consonants (abjads) (Daniels, 1990), or semantic meaning (logographies, *e.g.* Chinese characters) (Williams and Bever, 2010; Ding et al., 2004). We hypothesize that languages with more unique characters will have shorter words, as there are more unique sequences

Following basic combinatorial principles, with a larger set of unique characters, you can make more unique sequences (*e.g.* words) with shorter sequence length. English has 26 basic letters in its writing system. This means there are $26^2$ (676) possible two-letter words. For a language like Telugu, which has approximately 52 characters (Muppalaneni et al., 2019), therefore there are $52^2$ (2,704) possible two-letter words. Chinese has between 50,000 and 100,000 unique characters (The CLI Team, 2025), depending on the dictionary. However, the most common 3,500 characters make up over 99% of text seen in daily life (Hutong School, 2012). Therefore, Chinese may have 12,250,000 unique two-character words. And, indeed (Sun et al., 2018) 70% of the words in the Chinese Lexical Database (Sun et al., 2018) are two-character words.

---

[9]See Haspelmath (2017) for an overview.

We calculate the number of character unigrams (*i.e.* unique characters) and character bigram entropy. If a language has fewer unique characters, then we expect longer words according to the combinatorics described above. Therefore, we predict that the more unique characters a language has, the shorter its words will be. If a language has fewer unique characters, then we expect higher bigram entropy, because each character should be used in more unique contexts (bigrams) High bigram entropy, therefore, should be correlated with fewer unique characters and longer words. We calculate both metrics over the training corpus for our tokenizers for each language.

**Length Ratios.** We also take a measure of sequence length from the Byte Premium Tool (Arnett et al., 2024). One of the metrics released is the "length ratio", which is the ratio of number of characters needed to represent a text in one language relative to the number of characters needed to express parallel text in English. (also called called the "length ratio" (Arnett et al., 2024)). Higher length ratios corresponds to longer sequences.

Number of phonemes was the only metric relating to sequence length that did not significantly predict FLORES CTC. This is likely due to the arbitrary relationship between aspects of spoken languages and written language.

Length ratio is the most direct measurement of sequence length and explained the most variance. We tested whether unigram entropy or bigram entropy explain additional variance on top of length ratio. We fit a reduced linear mixed effects model with just length ratio as a fixed effect, and vocabulary size as a random intercept. We then fit two full models: one with length ratio and unigram entropy as fixed effects and one with length ratio and bigram entropy as fixed effects. We compare model fit with an ANOVA. Each unigram entropy and bigram entropy improve model fit (unigram entropy: $\chi^2(1) = 9.830, p = 0.002$; bigram entropy: $\chi^2(1) = 22.613, p < 0.001$). Furthermore, model with length ratio, unigram entropy, and bigram entropy is even better than just one with length ratio and unigram entropy ($\chi^2(1) = 82.736, p < 0.001$). In general, these effects are small, however.

## H.2   Script

We also wanted to know what the influence of script was on CTC. As mentioned at the end of Section 4, different writing systems encode different things. For instance, abjads are writing systems which represent each consonant with a symbol (Daniels, 1990), but vowels are often not represented. The scripts used for Arabic and Hebrew are examples languages that use abjads.

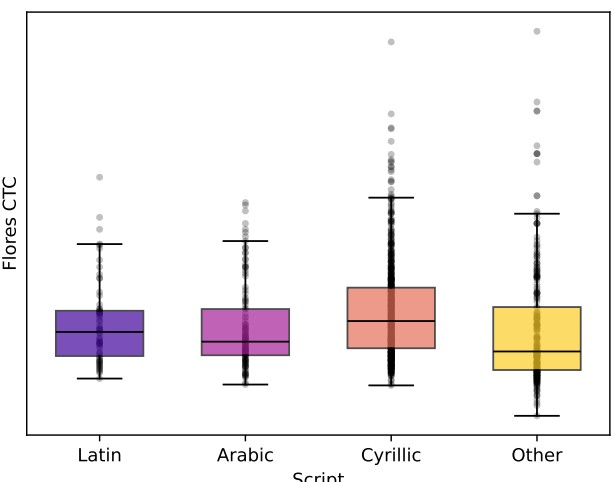

Figure 7: Boxplot showing CTCs across languages according to their ISO 15924 code, which indicates script type.

We first looked at scripts according to their ISO 15924 codes (Figure 7). Under this classification system, all scripts using a modified version of Latin script are classified as 'Latin'. The vast majority of the languages in our sample use Latin script. The only other scripts used by at least than three

languages were Arabic and Cyrillic. After grouping the languages using remaining scripts into an 'other' category, we found that script explains some variance ($R^2 = 0.109$ for the tokenizers with a vocabulary size of 65536). We report pairwise in Table 4. Latin script languages have significantly higher CTCs languages using Cyrillic or 'other' scripts. We believe this is likely another way that we can see the effect of the number of whitespace.

| Script 1 | Script 2 | $p$-value | Adjusted $p$-value | Significant |
|----------|----------|-----------|--------------------|-------------|
| Arabic | Cyrillic | 0.3560 | 2.1359 | False |
| Arabic | Latin | 0.0370 | 0.2222 | False |
| Arabic | Other | 0.0172 | 0.1033 | False |
| Cyrillic | Latin | $< 0.001$ | $< 0.001$ | **True** |
| Cyrillic | Other | 0.0319 | 0.1912 | False |
| Latin | Other | $< 0.001$ | $< 0.001$ | **True** |

Table 4: Pairwise Mann-Whitney U test results with Bonferroni correction. Significant comparisons (adjusted $p < 0.05$) are highlighted in bold.

Another possible way that we might see the effect of script is with the bytes-per-character (or, byte coefficient) metric. We measure this using the byte coefficients released as part of the Byte Premium Tool (Arnett et al., 2024). This metric, however, does not show any significant effects.

### H.3 Results for All Predictors at All Vocab Sizes

Tables 5, 6, and 7 report the results of individual linear correlations between these factors and FLORES CTC, as we did in Table 1. Some of these predictors are colinear and explain some of the same variance. We report these results so it is easier to interpret the amount of variance explained by each of the predictors.

| Vocab Size | 16384 | | 32768 | | 49152 | |
|------------|-----------|-------|-----------|-------|-----------|-------|
| **Predictor** | $p$-value | $R^2$ | $p$-value | $R^2$ | $p$-value | $R^2$ |
| n_phonemes | 0.543 | 0.004 | 0.601 | 0.003 | 0.747 | 0.001 |
| proportion_whitespace | 0.180 | 0.019 | **0.002** | **0.094** | **0.000** | **0.132** |
| unigrams_unique | **0.000** | **0.130** | 0.265 | 0.013 | 0.107 | 0.027 |
| bigrams_entropy_nospace | **0.011** | **0.066** | **0.043** | **0.042** | **0.010** | **0.067** |
| unigrams_entropy_nospace | **0.001** | **0.107** | 0.105 | 0.027 | **0.031** | **0.048** |
| char_coef | 0.825 | 0.001 | **0.004** | **0.085** | **0.003** | **0.089** |
| byte_coef | 0.126 | 0.024 | 0.125 | 0.025 | 0.107 | 0.027 |
| byte_premium | 0.126 | 0.024 | 0.125 | 0.025 | 0.107 | 0.027 |
| vocab_mean_token_len | **0.000** | **0.270** | 0.128 | 0.024 | 0.185 | 0.018 |
| flores_mean_token_len | **0.000** | **0.403** | **0.000** | **0.137** | **0.000** | **0.150** |
| data_sim | **0.000** | **0.315** | **0.000** | **0.242** | **0.000** | **0.193** |
| script | 0.683 | 0.042 | 0.953 | 0.128 | 0.828 | 0.138 |

Table 5: Significance is indicated with bolding.

| Vocab Size | 65536 | | 81920 | | 98304 | |
|---|---|---|---|---|---|---|
| **Predictor** | **$p$-value** | $R^2$ | **$p$-value** | $R^2$ | **$p$-value** | $R^2$ |
| n_phonemes | 0.843 | 0.000 | 0.906 | 0.000 | 0.951 | 0.000 |
| proportion_whitespace | **0.000** | **0.157** | **0.000** | **0.175** | **0.000** | **0.189** |
| unigrams_unique | 0.072 | 0.034 | **0.051** | **0.040** | **0.040** | **0.044** |
| bigrams_entropy_nospace | **0.005** | **0.079** | **0.003** | **0.089** | **0.002** | **0.096** |
| unigrams_entropy_nospace | **0.017** | **0.058** | **0.010** | **0.067** | **0.008** | **0.073** |
| char_coef | **0.003** | **0.088** | **0.003** | **0.088** | **0.003** | **0.087** |
| byte_coef | 0.090 | 0.030 | 0.080 | 0.032 | 0.073 | 0.033 |
| byte_premium | 0.090 | 0.030 | 0.080 | 0.032 | 0.073 | 0.033 |
| vocab_mean_token_len | 0.166 | 0.020 | 0.145 | 0.022 | 0.122 | 0.025 |
| flores_mean_token_len | **0.000** | **0.168** | **0.000** | **0.182** | **0.000** | **0.195** |
| data_sim | **0.000** | **0.239** | **0.000** | **0.296** | **0.000** | **0.325** |
| script | 0.692 | 0.144 | 0.612 | 0.148 | 0.559 | 0.151 |

Table 6: Significance is indicated with bolding.

| Vocab Size | 114688 | | 131072 | | 262144 | |
|---|---|---|---|---|---|---|
| **Predictor** | **$p$-value** | $R^2$ | **$p$-value** | $R^2$ | **$p$-value** | $R^2$ |
| n_phonemes | 0.989 | 0.000 | 0.981 | 0.000 | 0.871 | 0.000 |
| proportion_whitespace | **0.000** | **0.199** | **0.000** | **0.207** | **0.000** | **0.247** |
| unigrams_unique | **0.034** | **0.046** | **0.030** | **0.049** | **0.015** | **0.061** |
| bigrams_entropy_nospace | **0.002** | **0.100** | **0.001** | **0.104** | **0.000** | **0.123** |
| unigrams_entropy_nospace | **0.006** | **0.077** | **0.005** | **0.081** | **0.002** | **0.098** |
| char_coef | **0.004** | **0.085** | **0.004** | **0.084** | **0.005** | **0.079** |
| byte_coef | 0.066 | 0.035 | 0.061 | 0.036 | **0.043** | **0.043** |
| byte_premium | 0.066 | 0.035 | 0.061 | 0.036 | **0.043** | **0.043** |
| vocab_mean_token_len | 0.102 | 0.028 | 0.088 | 0.030 | **0.031** | **0.048** |
| flores_mean_token_len | **0.000** | **0.206** | **0.000** | **0.215** | **0.000** | **0.253** |
| data_sim | **0.000** | **0.347** | **0.000** | **0.426** | **0.000** | **0.442** |
| script | 0.522 | 0.154 | 0.497 | 0.156 | 0.414 | 0.164 |

Table 7: Significance is indicated with bolding.

# I  Parallel Data

## I.1  Plots

Figure 8 shows the CTCs for all seven outlier languages (Cebuano, Khmer, Luxembourgish, Mandarin Chinese, Maltese, Japanese, and Burmese) for tokenizers trained on parallel data.

## I.2  Statistical Tests for Parallel-Data Tokenizers

In Table 8, we report the results from the $t$-tests between the FLORES CTCs of the parallel and non-parallel tokenizers, by vocabulary size. None of the differences are significant.

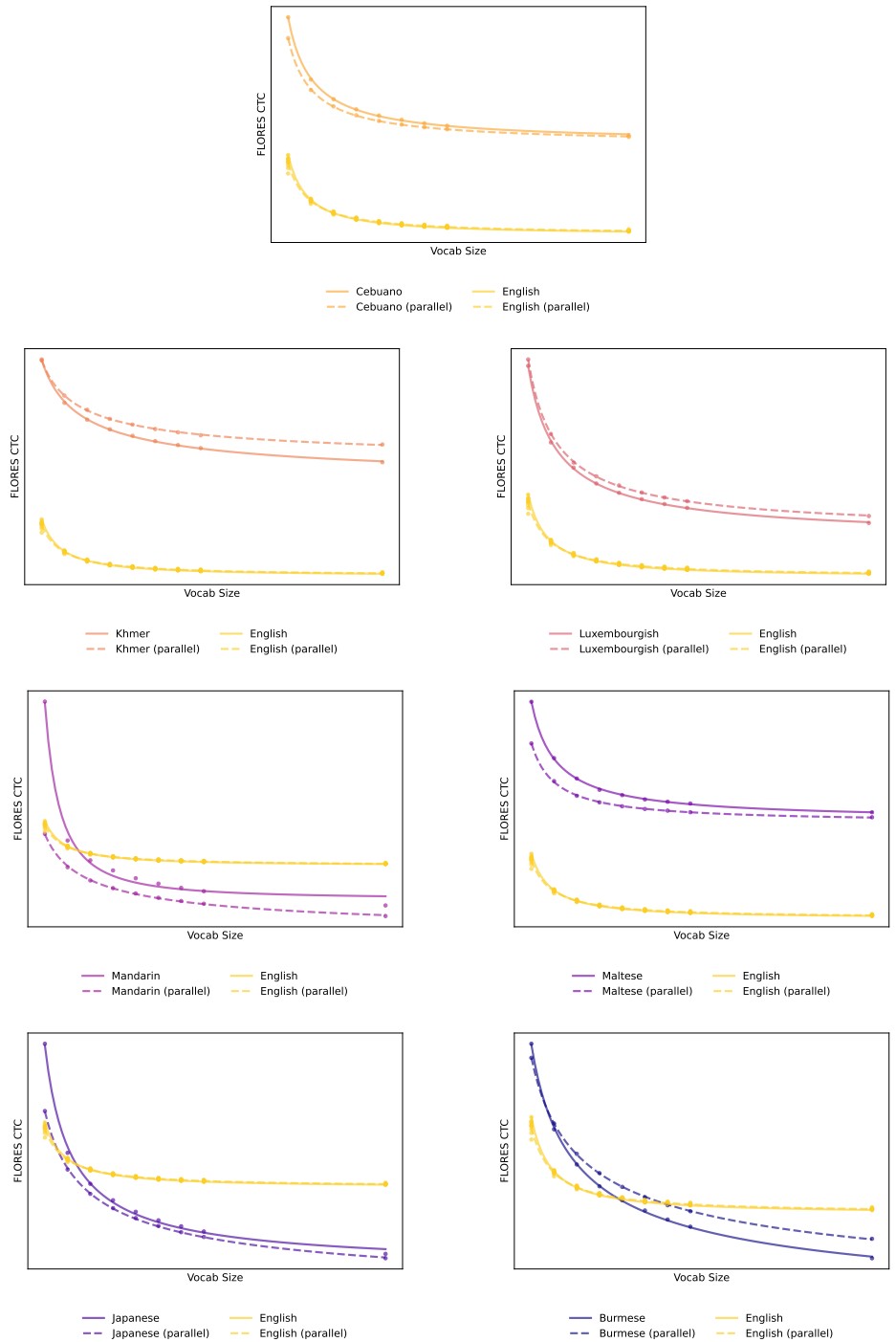

Figure 8: Cebuano, Khmer, Luxembourgish, Mandarin Chinese, Maltese, Japanese, Burmese

# J    Optimal-Vocabulary BPE Tokenizers

Here we describe our method for estimating optimal vocabularies, provide detail on training tokenizers with optimal vocabularies, and evaluate our estimation procedure.

| Vocab Size | $t$-Statistic | $p$-value | Mean Difference |
|---|---|---|---|
| 16384 | -2.075 | 0.054 | -5509 |
| 32768 | -1.862 | 0.081 | -787 |
| 49152 | -1.232 | 0.236 | -395 |
| 65536 | -0.777 | 0.449 | -225 |
| 81920 | -0.438 | 0.667 | -116 |
| 98304 | -0.338 | 0.739 | -85 |
| 114688 | -0.195 | 0.848 | -48 |
| 131072 | -0.034 | 0.973 | -8 |
| 262144 | 0.650 | 0.525 | 165 |

Table 8: $t$-test results and mean difference between parallel and non-parallel tokenizer CTCs for the seven languages in the sample. Significance is indicated with bolding.

### J.1 Estimating Optimal Vocabularies

In order to estimate the vocabulary size at which a tokenizer for a given language will reach a given CTC, we use the CTCs for the existing tokenizers and fit a power law curve, *e.g.* Figure 9.

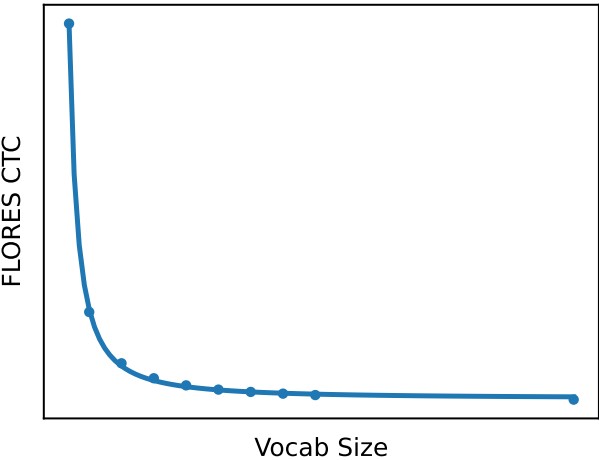

Figure 9: Power law relationship fit to the CTC values for the BP-unscaled BPE tokenizer for English.

The goal is to find the vocabulary size for each language at which the tokenizers are predicted to have the same CTC. For example, Figure 10 (left) shows the curves for English (`eng_latn`), Georgian (`kat_geor`), and Burmese (`mya_mymr`). There exists a vocabulary size for each language such that we predict a tokenizer would achieve a FLORES CTC of 52000 (indicated with the horizontal dashed line). However, this is not the case for all languages, *e.g.* Figure 10 (right). There is no vocabulary at which Telugu (`tel_telu`), Khmer (`khm_khmr`), and German (`deu_latn`) can achieve the same CTC.

Instead, for a range of CTCs (from 50000 to 74000 in increments of 1000) and estimated the vocabulary size at which every language would achieve the closest CTC to that target CTC. If our curve didn't predict a tokenizer could ever reach a target CTC, we used the CTC from the closest existing tokenizer. If the target CTC was below what we predicted could be achieved for a given language, we set the vocabulary size to the largest size we tested (262144) and set the predicted CTC to be the observed CTC at that size. If the prediction was higher than the worst CTC we observed, we set the vocab size to the lowest vocabulary size we tested (8192).

In Figure 11 we show the distribution of predicted CTCs for every target CTC. At the target CTC of 69000 there is the least amount of variance. This means if you wanted to select a vocabulary size at which most languages had a CTC closest together it would be approximately 69000. As target CTC gets lower, there are more languages which cannot achieve CTCs as low as the target and "get stuck"

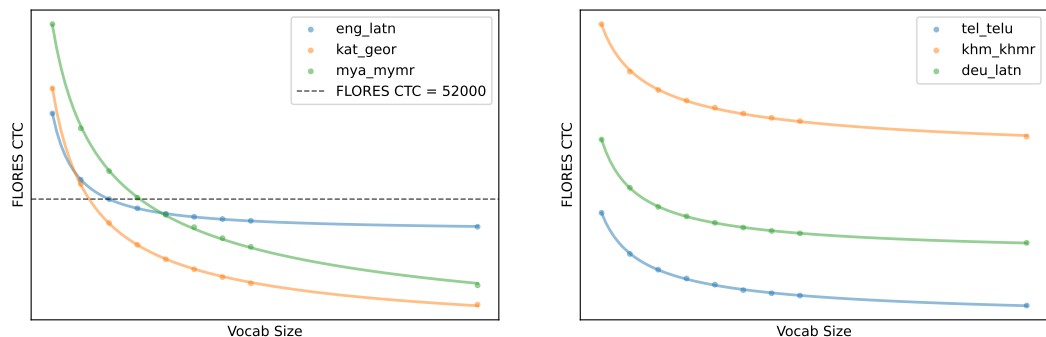

Figure 10: Left: CTCs across vocabulary sizes for English, Georgian, and Burmese. Horizontal black dashed line indicates a FLORES CTC of 52000. Intersections with the black line correspond to the "optimal" vocabulary size for each language. Right: CTCs for three languages, which do not have any vocabulary size at which all three languages can achieve the same CTC.

at a much higher CTC. For BPE tokenizers, there is a limit to the level of compression that can be achieved. This is reflected in the asymptote we see in the power law curves for each language.

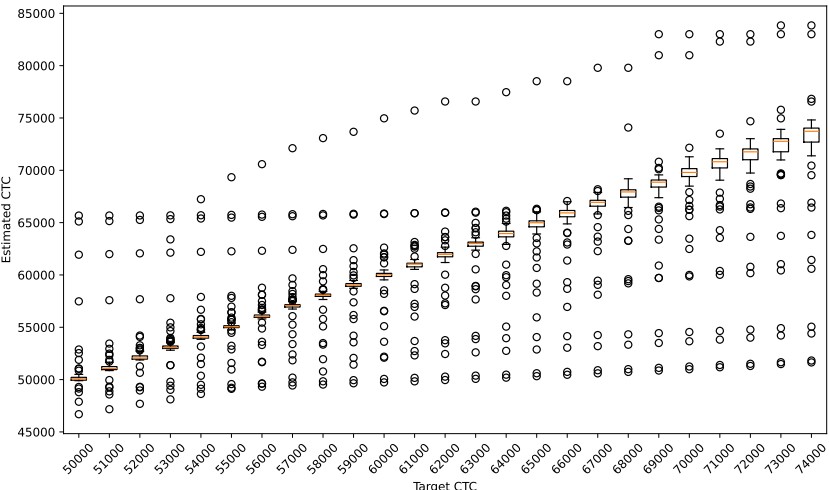

Figure 11: Each box plot shows the distribution of estimated CTCs for a given target CTC. In many cases, the distribution is so compressed that there is simply a gold line where the mean CTC is, and no box is visible.

## J.2 Tokenizer Training

For each target CTC, we trained the tokenizer with the vocabulary size that we estimated. If the vocabulary size was estimated at 8192 or 262144, we did not train a new tokenizer, and instead used the existing one with that vocabulary size. In total there are 2055 optimal-vocabulary tokenizers, 25 for each language. We used the same BPE implementation to train the tokenizers and trained them on the same training datasets.

## J.3 Estimated Vocab Validation

Next, we test how accurate our CTC estimation method was. We plot the distribution of CTCs for the estimated optimal-vocabulary tokenizers and actual optimal-vocabulary tokenizers in Figure 12, with the same-vocabulary tokenizers as reference. The estimated and actual optimal-vocabulary tokenizers have much less variance in their CTCs. The estimated CTCs are not perfectly accurate, but shows greatly reduced token premium effects compared to the same-vocabulary tokenizers.

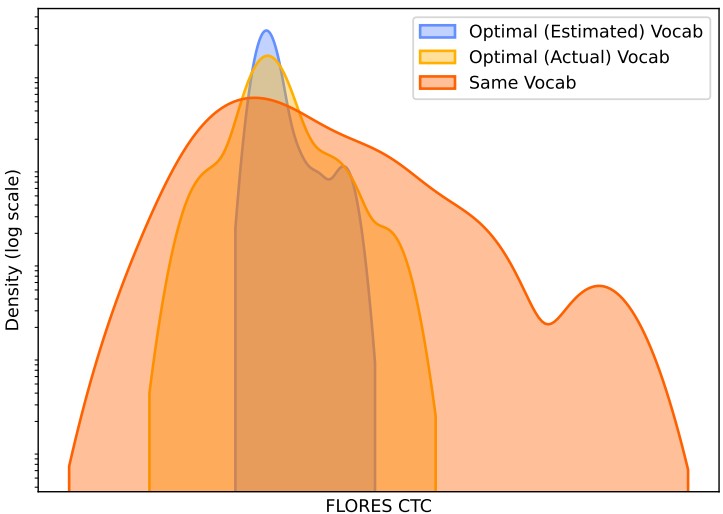

Figure 12: Density plot of same-vocab tokenizer CTCs, estimated optimal-vocab CTCs and actual optimal-vocab CTCs across languages. Y-axis is log-scaled to improve readability.

Overall, the RMSE between the actual and predicted values was 3733. We plot the distribution of CTCs across all languages in Figure 12. In Table 9, we report the RMSEs for each target CTC. As target CTC goes up, the RMSE goes down until CTC = 59000, then it increases monotonically.

| Target CTC | RMSE | Target CTC | RMSE |
|---|---|---|---|
| 50000 | 3536 | 60000 | 2913 |
| 51000 | 3185 | 61000 | 2992 |
| 52000 | 2975 | 62000 | 3059 |
| 53000 | 2938 | 63000 | 3144 |
| 54000 | 2982 | 64000 | 3283 |
| 55000 | 2933 | 65000 | 3445 |
| 56000 | 2921 | 66000 | 3563 |
| 57000 | 2880 | 67000 | 3758 |
| 58000 | 2870 | 68000 | 3981 |
| 59000 | 2870 | 69000 | 4384 |
|  |  | 70000 | 4601 |
|  |  | 71000 | 4892 |
|  |  | 72000 | 5083 |
|  |  | 73000 | 5350 |
|  |  | 74000 | 5651 |

Table 9: RMSE for estimated CTC method for each target CTC. RMSE decreases as target CTC increases on the left-hand column. Then, after 59000, RMSE increases as target CTC increases.

# K  SuperBPE Training

SuperBPE training works in two phases. In the first phase, the tokenizer trains like a normal BPE tokenizer. In the second half, the tokenizer learns merges over whitespace boundaries. In the first phase, the tokenizer learns a vocabulary of the desired size. Therefore, to learn superword tokens, you determine the number of tokens you wish to keep from the first phase, remove the rest, and then fill up the rest of the vocabulary in the second phase. The recommendation by the authors is to keep 180000 tokens from the first phase of training out of a total vocabulary size of 200000. The number of tokens retained from the first phase is called the *transition point*. In the original implementation of SuperBPE, they do not manipulate vocabulary size to the extent we do here. Therefore, in order to scale this appropriately to different vocabulary sizes, we set the transition point as a proportion of the full vocabulary size. Since the original recommendation was 90% of the full vocabulary, this was the highest transition point we tested. We also trained tokenizers with more aggressive transition points (see Section K.1 below), but in the paper all reported results are for tokenizers with a transition point of 0.9. We tested five different transition points: 0.5, 0.6, 0.7, 0.,8, and 0.9.

We trained SuperBPE tokenizers for all languages for all the vocabulary sizes we used before over 64000 (six sizes). In total, we train up to 2940 SuperBPE tokenizers. There were come cases where the training failed and additionally, we found some cases where there was an apparent error with byte fallback, which we describe in the following section.

The decrease in token premium effects is visualized in Figure 13. For a random sample of ten languages there is no vocabulary for each language such that a BPE tokenizer can achieve even approximately the same CTC. For SuperBPE, on the other hand, it is possible to achieve almost exactly the same CTC for the ten languages by training a tokenizer with each languages' optimal vocabulary size.

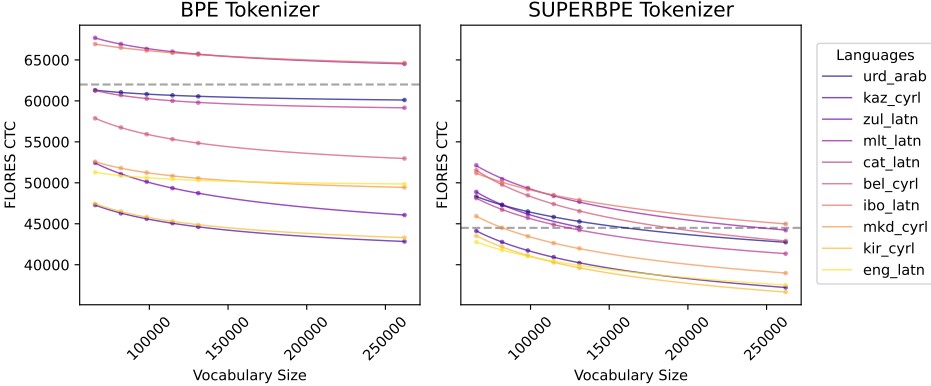

Figure 13: CTC curves for a random sample of ten languages for BPE and SuperBPE. This includes only tokenizers with vocabulary sizes greater than 64000.

## K.1  Transition Point Analysis

Here we report the distribution of CTCs by transition point. None of the transition points show greatly reduced token premium effects, however as transition point decreases, *i.e.* as the number of tokens retained from the first phase decreases, the compression trends downward. Figure 14 shows the distribution for each transition point.

# L  Additional SuperBPE Statistical Tests

We conduct an $F$-test to test whether there is a difference in the variance between the BPE and SuperBPE tokenizers. We also conduct a paired t-test to test whether the CTC is different between the two tokenizer types. All results are reported in Table 10. For all vocabulary sizes, SuperBPE tokenizers have less variance in CTC and have lower CTCs.

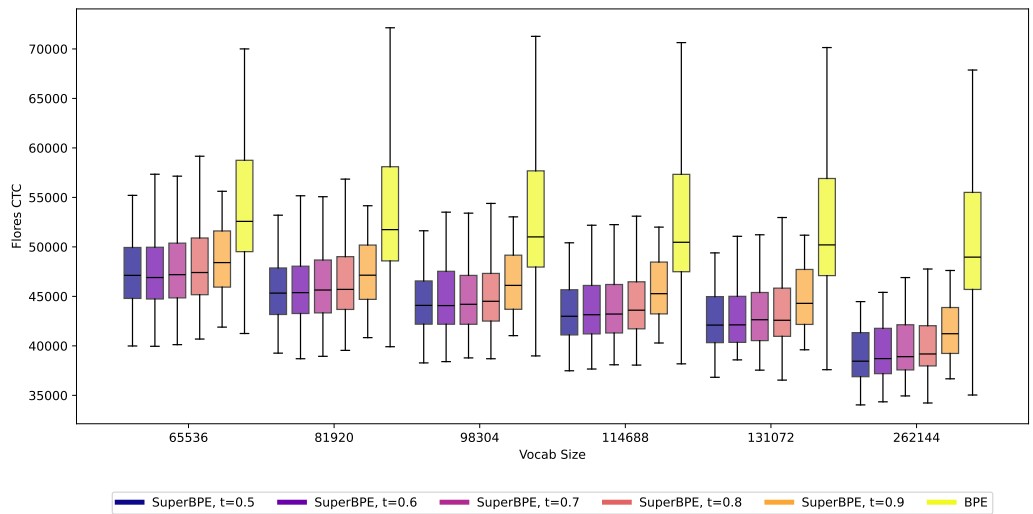

Figure 14: Distribution of CTCs by vocabulary size and transition point. These results do not include the outlier languages.

| Vocab Size | F-test | | | | t-test | |
|---|---|---|---|---|---|---|
| | $F$ | DF1 | DF2 | $p$-value | $t$ | $p$-value |
| 65536 | 2.187 | 73 | 96 | $< 0.001$ | -2.42 | 0.016 |
| 81920 | 2.006 | 70 | 96 | 0.002 | -2.67 | 0.008 |
| 98304 | 1.660 | 71 | 96 | 0.021 | -3.47 | $< 0.001$ |
| 114688 | 1.607 | 67 | 96 | 0.033 | -3.36 | $< 0.001$ |
| 131072 | 1.404 | 73 | 96 | 0.119 | -4.32 | $< 0.001$ |
| 262144 | 0.745 | 65 | 96 | 0.207 | -7.28 | $< 0.001$ |

Table 10: F-test and $t$-test results for SuperBPE tokenizers by vocabulary size.

Here we report the significance ($p$-value) and variance explained ($R^2$) for a linear regression predicting CTC from proportion of whitespaces for SuperBPE tokenizers. The relationship is not significant for any vocabulary size.

| Vocab Size | $R^2$ | Adj. $R^2$ | $p$-value |
|---|---|---|---|
| 65536 | 0.057 | 0.042 | 0.057 |
| **81920** | **0.086** | **0.070** | **0.023** |
| **98304** | **0.069** | **0.053** | **0.039** |
| **114688** | **0.079** | **0.062** | **0.034** |
| **131072** | **0.082** | **0.066** | **0.023** |
| **262144** | **0.111** | **0.094** | **0.012** |

Table 11: $R^2$ and $p$-value for regressions predicting CTC based on proportion of whitespaces for SuperBPE tokenizer with a threshold of 0.9.

## M  Optimal-Vocabulary SuperBPE Tokenizers

We use the same method to estimate CTCs for SuperBPE tokenizers as we did for BPE tokenizers (Appendix J). We plot the distribution of estimated CTCs for all target CTCs (Figure 15). The target CTCs at which there is near-perfect achievement of the same CTC across languages are much lower for SuperBPE tokenizers (48000 to 57000) than for BPE (69000). And there is a larger range of CTCs for which it is possible to get a tokenizer for every language to achieve.

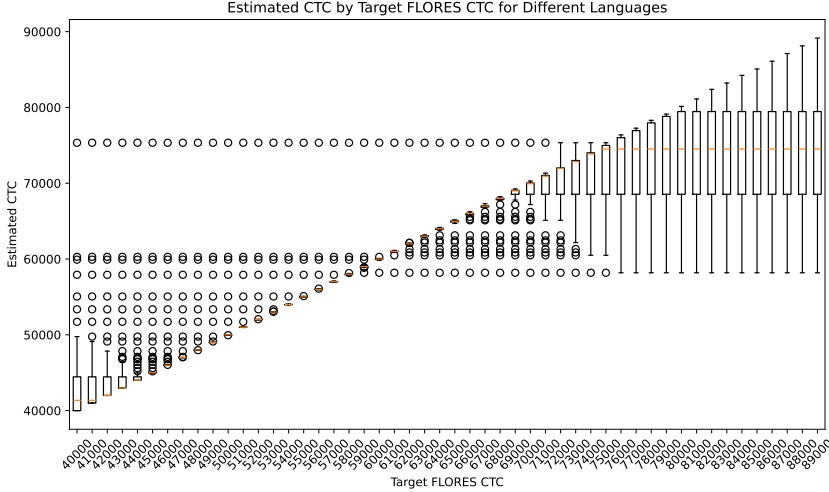

Figure 15: Distribution of estimated CTCs for each target CTC for SuperBPE tokenizers.

# N    Remaining Variance

We fit a linear mixed effects model with unigram entropy, bigram entropy, length ratio (character coefficient), and bytes-per-character ratio (byte coefficient) as predictors. We included vocabulary size as a random intercept. The results of the ANOVA are reported in Table 12.

| Predictor | SS | MS | df (num) | df (den) | F | p-value |
|---|---|---|---|---|---|---|
| Unigram entropy | 10,084,887 | 10,084,887 | 1 | 356.05 | 0.68 | .411 |
| Bigram entropy | 1,031,215 | 1,031,215 | 1 | 356.06 | 0.07 | .793 |
| Length ratio | **185,829,302** | **185,829,302** | **1** | **356.00** | **12.48** | **< .001** |
| Byte coef | **68,017,284** | **68,017,284** | **1** | **356.01** | **4.57** | **.033** |

Table 12: ANOVA results for each of the fixed effects

