# OpenReview forum: "Explaining and Mitigating Crosslingual Tokenizer Inequities"
_NeurIPS.cc/2025/Conference — NeurIPS 2025 poster_

### Official Review · Reviewer_kKAc · 2025-06-23

**Clarity:** 3
**Significance:** 2
**Originality:** 3
**Rating:** 3
**Confidence:** 5

**Summary:**

This paper examines the phenomenon of cross-lingual tokenization inequalities, i.e., the observation made in prior work that tokenizers generate significantly longer token sequences for some languages compared to others. To investigate this, the authors train a large number of monolingual tokenizers, controlling for various parameters (e.g., tokenization algorithm, vocabulary size), and analyze their properties. They find that the similarity between tokenizer training and test data does not impact tokenization inequalities, but vocabulary size and pre-tokenization do, thus pointing to potential mitigation strategies.

**Questions:**

- Do your findings translate to better performance in terms of language modeling and downstream tasks?
- Can you provide qualitative analyses to better explain your quantitative findings?

**Ethical Concerns:**

["NO or VERY MINOR ethics concerns only"]

**Final Justification:**

After the rebuttal, I still see critical limitations, so I believe the score of 3 is justified.

**Limitations:**

The authors do not discuss the potential negative societal impact of their work. In terms of general limitations, the authors should mention that they do not analyze language model performance, which in my view is a key limitation if this paper.

**Paper Formatting Concerns:**

None.

**Quality:**

2

**Strengths And Weaknesses:**

The paper provides a useful extension of prior work on cross-lingual tokenization inequalities. The finding that these disparities are not (solely) caused by the training data is interesting, as are the other analyses provided by the authors (e.g., that different languages need to be allocated different vocabulary sizes to achieve comparable compression). Furthermore, the 7,000 tokenizers trained and released by the authors will benefit future research on tokenization in general, and tokenization fairness in particular.

While these contributions are laudable, I have two main concerns:
- The paper only analyzes statistical properties of the tokenizers but does not examine how these properties affect language models trained with them. Fairness in terms of performance is a key motivating factor of prior work on tokenization inequalities (e.g., see §4.3 in [Ahia et al., 2023](https://aclanthology.org/2023.emnlp-main.614/)), so this issue, I think, is very much in-scope for the current paper. For example, it is unclear whether the ideal vocabulary sizes in terms of token premiums also result in the best relative performance compared to English.
- The analyses provided by the authors stay on a somewhat superficial level, mostly because the authors focus on quantitative results. It seems important to also provide _qualitative_ analyses that back up the quantitative findings. For example, linguistic, and in particular morphological, analyses might provide for a better understanding of the quantitative findings.

Minor points:
- The SentencePiece analysis in §3.1 is unclear. SentencePiece is a library that implements both Unigram and BPE, so is this analysis about implementation differences between HuggingFace and SentencePiece? It seems the authors just compared their tokenizers with the tokenizers trained for the Goldfish models, but this means that practically all aspects of tokenizer training are different, so it is difficult to tell how much we can learn from the comparison.
- The authors missed a key related work, specifically a NeurIPS paper from last year that proposed a new method to mitigate cross-lingual tokenization inequalities ([Ahia et al., 2024](https://openreview.net/forum?id=1e3MOwHSIX)).

---

> ### Author Rebuttal · Authors · 2025-07-31
>
> >The paper only analyzes statistical properties of the tokenizers but does not examine how these properties affect language models trained with them. Fairness in terms of performance is a key motivating factor of prior work on tokenization inequalities (e.g., see §4.3 in Ahia et al., 2023), so this issue, I think, is very much in-scope for the current paper. For example, it is unclear whether the ideal vocabulary sizes in terms of token premiums also result in the best relative performance compared to English.
>
> First, we think that the statistical properties of the tokenizers are important in and of themselves. Tokenization affects the cost of training and inference, as it determines the number of computations over a given text or prompt. We will emphasize these points in the updated paper. We agree that it is also important to follow up these findings by linking these results to model performance on specific tasks. The aim of this paper is to offer broad coverage of languages, tokenization algorithms, and vocabulary sizes. Therefore, we are not able to train models for all of the tokenizers we train, due to compute limitations. However, we hope that future work can test whether these findings also link to differences in language model performance.
>
> >The analyses provided by the authors stay on a somewhat superficial level, mostly because the authors focus on quantitative results. It seems important to also provide qualitative analyses that back up the quantitative findings. For example, linguistic, and in particular morphological, analyses might provide for a better understanding of the quantitative findings.
>
> We agree that it would be interesting to investigate other aspects of these tokenizers. Morphological alignment, for example, is difficult to evaluate comparably across languages, especially languages with very little overt morphology, e.g. Chinese, or languages with non-concatenative morphology, e.g. Arabic (see [2] for discussion on this point), which poses theoretical and methodological challenges for addressing this question. We release our tokenizers, which enables others to investigate this in the future.
>
> >The SentencePiece analysis in §3.1 is unclear. SentencePiece is a library that implements both Unigram and BPE, so is this analysis about implementation differences between HuggingFace and SentencePiece? It seems the authors just compared their tokenizers with the tokenizers trained for the Goldfish models, but this means that practically all aspects of tokenizer training are different, so it is difficult to tell how much we can learn from the comparison.
>
> Yes, we agree that we were unclear on this point. Here, SentencePiece refers to the SentencePiece implementation of Unigram, which contrasts with the HuggingFace implementation. As we show, the two offer very different results, despite being based on the same algorithm. This is likely due to implementation differences. We will clarify these points in the updated paper.
>
> >The authors missed a key related work, specifically a NeurIPS paper from last year that proposed a new method to mitigate cross-lingual tokenization inequalities (Ahia et al., 2024).
>
> We thank the reviewer for highlighting this paper. One difference between our method and that in Ahia et al. (2024) is that our approach does not require modifying tokenization algorithms or model architectures. We use existing tokenization algorithms, and only manipulate vocabulary size. We will include further discussion of the Ahia paper and how it differs from our approach in the updated paper.
>
> >Do your findings translate to better performance in terms of language modeling and downstream tasks?
>
> This was not evaluated in the current work, but we agree that addressing this would be very valuable!
>
> >Can you provide qualitative analyses to better explain your quantitative findings?
>
> We provided some qualitative discussion in Appendix F, but will try to include more illustrative examples in the updated manuscript to clarify the points in the paper!
>
> >The authors do not discuss the potential negative societal impact of their work. In terms of general limitations, the authors should mention that they do not analyze language model performance, which in my view is a key limitation if this paper.
>
> We do not foresee any negative societal impact of the work. In fact, we hope that the outcome of the paper is improved crosslingual equity in tokenization and language models more generally. We agree that analyzing language model performance would strengthen the work, and will include discussion of this in the Limitations section.
>
>
> [1] Kreutzer, J., Caswell, I., Wang, L., Wahab, A., Van Esch, D., Ulzii-Orshikh, N., ... & Adeyemi, M. (2022). Quality at a glance: An audit of web-crawled multilingual datasets. Transactions of the Association for Computational Linguistics, 10, 50-72.
>
> [2] Catherine Arnett, Marisa Hudspeth, and Brendan O’Connor (2025). Evaluating Morphological Alignment of Tokenizers in 70 Languages. Tokenizer Workshop at ICML. Vancouver, Canada.

---

> > ### Comment · Reviewer_kKAc · 2025-08-01
> >
> > I thank the authors for their rebuttal.
> >
> > I continue to believe that evaluating on language modeling and/or downstream tasks is important. I appreciate the authors' comment about this, and I agree that doing so for all tokenizers is infeasible, but at least a small-scale evaluation on a few strategically picked languages would be desirable.
> >
> > I also continue to believe that an in-depth qualitative analysis is missing in the paper; appendix F does not really provide a qualitative discussion as claimed in the rebuttal but merely lists examples from the FLORES dataset.
> >
> > Due to these remaining limitations, I will keep my score at 3.

---

### Official Review · Reviewer_ehmG · 2025-06-29

**Clarity:** 3
**Significance:** 3
**Originality:** 3
**Rating:** 4
**Confidence:** 4

**Summary:**

The paper studies how tokenization unfairness arises. It trains several tokenizers (BPE, Unigram, and SuperBPE) and uses corpus token count (CTC) to measure the factors that impact the “premium” cost paid by under-represented languages by LLMs. The identified vocabulary size and the pre-tokenizer are the most impactful factors to reduce unfairness.

**Questions:**

More like a remark: I personally suspect that the experiments on FLORES do not allow for capturing how tokenizers work in real case scenarios. In other words, the unfairness emerges as LLMs use tokenizers mostly trained to work with English on other high-coverage languages (e.g., in a BPE multi-lingual tokenizer, most merges are for English, while other languages remain closer to character-level tokenization).

Do you think that, for a language with high CTC, if we could train a tokenizer on the same amount of data as the English counterpart, we would achieve a significant reduction in the premium? This seems to me an experiment one can try with an LLM mostly trained on a massive non-English corpus (e.g., Chinese) and check if the premium for English is high.

Line 294, “Perhaps too much compression could be giving the model effectively less time to think, by dedicating fewer FLOPS to predicting 295 each next token.”. This does not seem reasonable to me. How does this hypothesis explain why English LLMs are better than (mostly) any other language? If a token appears so frequently that it is merged, it probably means that its “meaning” can be compressed and that the information in the sentence lies somewhere else.

**Ethical Concerns:**

["NO or VERY MINOR ethics concerns only"]

**Final Justification:**

I think this paper has some merits and I have nothing against seeing it at the conference.

**Limitations:**

The paper addresses a relevant problem and conducts the experiments well. I do not see any limitation, given what they claim as a contribution.

One limitation is that FLORES-200 is not a very large dataset (it contains 3K sentences per-language) and may still be biased (though less than the previous version) towards specific high-resource languages such as English, Spanish, etc.

**Paper Formatting Concerns:**

No concerns.

**Quality:**

3

**Strengths And Weaknesses:**

**Strengths**

Tokenization unfairness is a problem that was recently identified as relevant by the LLMs research community. While most of the literature, including the paper that identifies the issues, does not introduce mitigations, this paper addresses them.

Fig. 1 and the relative sentence, “Monolingual tokenizers trained with exactly the same implementation, dataset size, and vocabulary size demonstrate widely variable token premium effects.” This is an interesting finding that contradicts one of the hypothesis in (Petrov et al 2023) that unfairness may disappear with larger multi-lingual training corpora.

Line 245: “Tokenizers with optimal vocabulary sizes per language have reduced token premium effects compared to tokenizers with the same vocabulary size across languages”. I personally find this to be the best contribution of the paper, as it is a step towards mitigating tokenizers unfairness.

Line 283: “there is no vocabulary size at which monolingual tokenizers will achieve similar compression across a wide variety of languages”. This seems a trivial observation, but it is not and tells us we cannot rely on the same tokenization technique (and parameters) for any language, if we aim to achieve tokenization fairness.


**Weaknesses**

The FLORES-200 dataset is small (in the sense that it does not contain too many examples per language). On the other hand, I understand there is a scarcity of multi-lingual corpora, so this is a weakness of the current state of the art in multi-lingual NLP and not of your paper (but results may be affected by this limitation).

I am not sure I agree with all the findings the authors describe in the paper. For example, line 303, “Our work shows that some languages need larger vocabularies than others for comparable compression.”. Is that a finding that is correlated with the nature of FLORES, which starts from sentences in English (mostly, and some Spanish and a few other languages, if I remember correctly), and provides translations for them? Would the results change if there were a FLORES whose original sentences are (e.g.,) more prototypical Chinese sentences (e.g., so that the English translation should adapt to them)?

---

> ### Author Rebuttal · Authors · 2025-07-31
>
> >The FLORES-200 dataset is small (in the sense that it does not contain too many examples per language). On the other hand, I understand there is a scarcity of multi-lingual corpora, so this is a weakness of the current state of the art in multi-lingual NLP and not of your paper (but results may be affected by this limitation).
>
> We agree that this is a limitation of multilingual NLP, but we are not aware of any larger multi-parallel datasets for this many languages.
>
> >I am not sure I agree with all the findings the authors describe in the paper. For example, line 303, “Our work shows that some languages need larger vocabularies than others for comparable compression.”. Is that a finding that is correlated with the nature of FLORES, which starts from sentences in English (mostly, and some Spanish and a few other languages, if I remember correctly), and provides translations for them? Would the results change if there were a FLORES whose original sentences are (e.g.,) more prototypical Chinese sentences (e.g., so that the English translation should adapt to them)?
>
> Given that the compression differences between languages are so extreme in many cases, it seems unlikely that any potential biases introduced by translation can account for such large effects, but we agree that this is an important consideration.
>
> >I personally suspect that the experiments on FLORES do not allow for capturing how tokenizers work in real case scenarios. In other words, the unfairness emerges as LLMs use tokenizers mostly trained to work with English on other high-coverage languages (e.g., in a BPE multi-lingual tokenizer, most merges are for English, while other languages remain closer to character-level tokenization).
>
> The experiments in this paper will not capture the biases observed in existing multilingual tokenizers, which indeed often focus on English. In addition to the differences we observe in the paper, we also expect interactions between languages (e.g. due to shared vs. different writing systems and linguistic similarity) to play roles. In future work, we hope to better understand these dynamics in a multilingual context.
>
> >Do you think that, for a language with high CTC, if we could train a tokenizer on the same amount of data as the English counterpart, we would achieve a significant reduction in the premium? This seems to me an experiment one can try with an LLM mostly trained on a massive non-English corpus (e.g., Chinese) and check if the premium for English is high.
>
> With respect to the first part of this question, we compared our tokenizers to ones trained on much more data. The results suggest that tokenizer training data size alone may not lead to lower CTC (see the paragraph starting on L117 and Appendix C). With respect to the second part of this question, we understand the reviewer to be asking about compression in a multilingual setting. In this paper, we focus on the monolingual setting. We observe that tokenizer compression does not generalize or transfer well to new languages, but in future work, we hope to evaluate the interactions between languages in multilingual tokenizers. We would appreciate clarification from the reviewer as to whether we have addressed your questions!
>
> >Line 294, “Perhaps too much compression could be giving the model effectively less time to think, by dedicating fewer FLOPS to predicting 295 each next token.”. This does not seem reasonable to me. How does this hypothesis explain why English LLMs are better than (mostly) any other language? If a token appears so frequently that it is merged, it probably means that its “meaning” can be compressed and that the information in the sentence lies somewhere else.
>
> Compression is not the only factor that explains why language models may perform better in English than for other languages. In most cases, models perform best in English because they have been primarily trained to optimize performance for English. They are trained on mostly English data. English data quality is usually also higher (see [1]). Furthermore, we do not find that English has the best compression in the monolingual setting. Therefore, we would not predict that all else being equal (including dataset quality) in a monolingual setting, that English would necessarily have the best performance.
>
> >One limitation is that FLORES-200 is not a very large dataset (it contains 3K sentences per-language) and may still be biased (though less than the previous version) towards specific high-resource languages such as English, Spanish, etc.
>
> We agree that this is a concern. This is the motivation for the experiments in Section 5.1, in which we control for domain similarity and data quality. The results suggest that this does not significantly affect the compression results in this paper.
>
> [1] Kreutzer, J., Caswell, I., Wang, L., Wahab, A., Van Esch, D., Ulzii-Orshikh, N., ... & Adeyemi, M. (2022). Quality at a glance: An audit of web-crawled multilingual datasets. Transactions of the Association for Computational Linguistics, 10, 50-72.

---

> > ### Comment · Reviewer_ehmG · 2025-08-01
> >
> > I am happy with the reviewers response, so I will keep my score.

---

### Official Review · Reviewer_SFsu · 2025-07-03

**Clarity:** 4
**Significance:** 3
**Originality:** 3
**Rating:** 4
**Confidence:** 4

**Summary:**

This paper presents a large-scale empirical investigation of "token premiums", the phenomenon where encoding the same content requires a different number of tokens depending on the language. The authors systematically study this issue in a monolingual context to isolate it from the confounding factor of data imbalance in multilingual tokenizers. By training nearly 7,000 tokenizers across 98 languages while controlling for vocabulary size, dataset size, and tokenizer algorithm (BPE, Unigram), they demonstrate that token premium inequities persist even under these controlled conditions.

**Questions:**

- Why not taking the "best" and "worst" (and maybe some in between) tokenizers based on your metrics for a few languages which had tasks available and train some MLM or small LLM and perform downstream tasks that allow to get an idea of the performances gap, between what we can consider a "good" or "bad" tokenizer after reading this paper ? Languages will not be comparable between each others, but at least we will get an idea of the impact of the tokenizer optimality.

**Ethical Concerns:**

["NO or VERY MINOR ethics concerns only"]

**Final Justification:**

The discussion with the authors clarified some aspects of the paper.

**Limitations:**

- The paper compellingly demonstrates how to achieve tokenization equity in a monolingual context. The natural extension is to consider the common scenario of a single, unified tokenizer for multilingual text, code, and mathematical formulas. Do the authors have a view on how their findings might translate to this challenge? Given that any unified tokenizer is arguably sub-optimal by design, have the authors considered whether a "mixture-of-tokenizers" might be a more promising direction? For instance, a system could dynamically route inputs to different language or domain-specific tokenizers, building directly on this paper's findings to mitigate premiums in complex, mixed-data settings.

**Quality:**

2

**Strengths And Weaknesses:**

Strengths:
- The scale and systematic nature of this study are its greatest strengths. Training ~7,000 tokenizers across 98 languages is a massive undertaking that provides robust empirical evidence. To my knowledge, this is the first study to systematically disentangle token premiums from multilingual training effects by focusing on a controlled monolingual setup.
- Benchmarking against established, publicly-available tokenizers provided a valuable real-world baseline for the compression metrics.
- The findings have immediate practical implications for anyone building multilingual models, offering clear strategies (SuperBPE, optimal vocabulary sizing) to build fairer and more cost-effective systems.
- The paper is well-written and structured. It logically moves from establishing the problem (Section 1-3), to investigating its causes (Section 4), and finally to testing solutions (Section 5). The figures, particularly the density plots, are effective at visualizing the core problem (the spread of CTCs) and the success of the proposed interventions.

Weaknesses:
- My primary concern, is the overlap between the training and evaluation data. The paper uses FLORES-200 for evaluation as mentioned L. 128, but this same dataset is listed as a training source in the accompanying repository for the data collection used tylerachang/curse-of-multi-linguality on GitHub (L. 827). This data leakage means the tokenizers were evaluated on familiar data, likely leading to an overestimation of their true compression performance on unseen text. This issue significantly weakens the paper’s empirical claims. The authors must address this by either demonstrating that the specific FLORES-200 splits used for evaluation were properly excluded from training or by re-evaluating their trained tokenizers on a completely novel, held-out dataset.
- The recommended "best" solution, SuperBPE, fails to train for 14 languages (L. 257), particularly those using South and Southeast Asian scripts. This limits the universality of the proposed solution, even if the results were reliable.

---

> ### Author Rebuttal · Authors · 2025-07-31
>
> >My primary concern, is the overlap between the training and evaluation data.
>
> We thank the reviewer for highlighting this point. We removed FLORES from the training datasets before training the tokenizers, and we will note this in the updated paper. Additionally, since submitting the manuscript, we conducted a contamination analysis and found that for every language except Samoan, there was a maximum of 1 item from FLORES in our training dataset. Samoan was significantly contaminated. We will address the contamination and re-train the Samoan tokenizers, and update the results.
>
> >The recommended "best" solution, SuperBPE, fails to train for 14 languages (L. 257), particularly those using South and Southeast Asian scripts. This limits the universality of the proposed solution, even if the results were reliable.
>
> Since submitting the paper, the authors of the SuperBPE paper have helped us identify and address the solution. In fact, the issue is related to the decoder, which can be fixed without re-training. We will include the previously excluded languages in the analysis in the updated manuscript.
>
> >Why not taking the "best" and "worst" (and maybe some in between) tokenizers based on your metrics for a few languages which had tasks available and train some MLM or small LLM and perform downstream tasks that allow to get an idea of the performances gap, between what we can consider a "good" or "bad" tokenizer after reading this paper ? Languages will not be comparable between each others, but at least we will get an idea of the impact of the tokenizer optimality.
>
> In this paper, we focus only on intrinsic tokenizer evaluation. We agree with the reviewers that downstream evaluations would be informative, and think this is a valuable direction for future work.
>
> >The paper compellingly demonstrates how to achieve tokenization equity in a monolingual context. The natural extension is to consider the common scenario of a single, unified tokenizer for multilingual text, code, and mathematical formulas. Do the authors have a view on how their findings might translate to this challenge? Given that any unified tokenizer is arguably sub-optimal by design, have the authors considered whether a "mixture-of-tokenizers" might be a more promising direction? For instance, a system could dynamically route inputs to different language or domain-specific tokenizers, building directly on this paper's findings to mitigate premiums in complex, mixed-data settings.
>
> Indeed, we hope to extend these findings to multilingual settings. One way in which the findings from the monolingual settings inform work on multilingual settings is that they predict that a multilingual tokenizer should dedicate different proportions of the vocabulary to different languages in order to achieve balanced compression. However, other factors such as script overlap and linguistic similarity will also likely play important roles. We hope to conduct experiments on this in the near future and create more optimal multilingual tokenizers. We thank the reviewer for these suggestions!

---

### Official Review · Reviewer_TBJw · 2025-07-03

**Clarity:** 3
**Significance:** 1
**Originality:** 3
**Rating:** 4
**Confidence:** 3

**Summary:**

This paper investigates token premiums, the extra tokens required to encode text in different languages. By training ~7k tokenizers across 98 languages, the authors find that vocabulary size and pre-tokenization drive these differences. They also investigate Superword tokenizers and find that they reduce token premiums and improve compression.

**Questions:**

I do not have any questions. The weaknesses are mostly inherent to the topic and its significance. I scored 3 for clarity because the paper is clear enough; however, I think the significance is limited (score 1). The originality is good (score: 3) but not excellent, as the CTC concept has been previously introduced and explored. Overall, the quality is good (score 3).


Suggestions:
- The main suggestions are included in the weaknesses section. You could solve them. OR perhaps if i misunderstood inform me of what do you think is right. These rest of suggestions here are just extra and do not impact your score.
- In Figure 1 (page 2), the term CTC is introduced without explanation, and only later (on page 3) is it clarified it stands for. Add the acronym in both the text and figure caption for better readability.
- You could try using more diverse parallel datasets beyond FLORES. I think most paper just use FLORES for token premiums. But for example, UDHR (as shown in Figure 3 of https://arxiv.org/abs/2309.13320) covers additional scripts not present in FLORES. You might also find useful parallels in the [MTEB multilingual bitext mining task](https://github.com/embeddings-benchmark/mteb/tree/main/mteb/tasks/BitextMining/multilingual). I suggest this because your paper's main contribution is extensive comparison; adding this dimension could further deepen your comparisons.

**Ethical Concerns:**

["NO or VERY MINOR ethics concerns only"]

**Final Justification:**

I will keep my positive score the same as before.

**Limitations:**

yes

**Paper Formatting Concerns:**

I did not notice any major formatting issues.

**Quality:**

3

**Strengths And Weaknesses:**

Strengths:
- To the best of my knowledge, this is the first large-scale systematic study of token premiums across nearly 7k monolingual tokenizers covering 98 languages.
- The paper offers actionable insights, showing that factors such as vocabulary size, whitespace-agnostic tokenization (SuperBPE), and pre-tokenization strategies can significantly influence token premiums across languages.

Weaknesses:
- This is not necessarily a weakness of this paper, but a limitation of the broader research area: token premiums may not be highly important in practice. At inference time, users tend to use high-resource languages, so the token inefficiency in low-resource languages—while unfair—may have limited real-world impact. The paper does not account for this.
- The main metric used is CTC, and the evaluation is based on the FLORES dataset, which is parallel across languages. Both the metric and dataset have already been extensively used in prior work—Ahia et al. (2023) and Petrov et al. (2023)—which inspired this study. As a result, this paer is not new and the strength is only about the high number of the comparisons.
- The study focuses only on tokenizer experiments, much like other tokenizer studies it does not  evaluate whether these improved tokenization settings actually lead to better model performance. This limits the practical impact of the findings. It's possible that an "optimal" tokenizer in terms of paper performs worse during downstream tasks after finishing the model training, but this is never tested.
- The paper is multilingual and comparative, but not a "crosslingual" paper per se; the term "crosslingual" appear 4 times: one in title: one in abstract; one in conclusion. I'm pretty sure the paper title needs a change.

---

> ### Author Rebuttal · Authors · 2025-07-31
>
> >a limitation of the broader research area: token premiums may not be highly important in practice. At inference time, users tend to use high-resource languages, so the token inefficiency in low-resource languages—while unfair—may have limited real-world impact.
>
> One of the goals of this work is to work towards greater crosslingual equity for lower-resource languages (which by definition impacts fewer users), and the impact of this work is significant for members of those language communities. In this paper, we consider only monolingual tokenizers. If we were to apply these findings to language models, we would expect that these models would also be monolingual, and therefore at inference time, only the efficiency of the target language should be considered. However, in the future, we hope to adapt these findings to multilingual contexts, in which case we would want to balance the efficiency of several languages.
>
> >The main metric used is CTC [...] Both the metric and dataset have already been extensively used in prior work [...] As a result, this paper is not new and the strength is only about the high number of the comparisons.
>
> We use the same method as Ahia et al. (2023) and Petrov et al. (2023), however in those papers, the authors study existing multilingual tokenizers. Neither of those papers studies monolingual tokenizers, nor do they train their own tokenizers or manipulate factors such as vocabulary size. The contribution of this paper is demonstrating that identical training in a monolingual context still leads to differences across languages, which to our knowledge has not been shown before. Understanding how to mitigate these differences is an essential step in reducing inequities in multilingual contexts.
>
> >The study focuses only on tokenizer experiments, much like other tokenizer studies it does not evaluate whether these improved tokenization settings actually lead to better model performance. This limits the practical impact of the findings.
>
> We agree that it would be potentially valuable to train models for each of the tokenizers to understand how differences in tokenizers impact performance. We do not currently have the resources to train at least 7,000 models. The contribution of the paper is to understand the relationship between language features and intrinsic features of tokenizers, which before now has not been demonstrated. Future work could train a small number of models to test whether using SuperBPE tokenizers with optimal vocabulary sizes leads to better performance, but this is outside the scope of the research questions set out in this paper.
>
> > The paper is multilingual and comparative, but not a "crosslingual" paper per se
>
> In the paper, we only train and evaluate monolingual tokenizers, therefore the paper does not address multilingual tokenization (i.e. a single tokenizer trained on multiple languages). We use crosslingual to refer to comparisons across languages in monolingual settings. We will clarify our usage of this terminology in the updated version.
>
> >the term CTC is introduced without explanation
>
> Thank you for highlighting this, we will address this in the updated version!
>
> >You could try using more diverse parallel datasets beyond FLORES.
>
> We thank the reviewer for the suggestion. Arnett et al. [1] compared FLORES and UDHR (and also the Bible) as parallel corpora and found that UDHR was both smaller and noisier than FLORES (because, for example, there are many domain-specific and borrowed terms). But it is possible that UDHR could be good to investigate for future expansions of this work. We will include discussion on this point in the updated manuscript.
>
> [1] Arnett, C., Chang, T. A., & Bergen, B. (2024). A Bit of a Problem: Measurement Disparities in Dataset Sizes across Languages. In Proceedings of the 3rd Annual Meeting of the Special Interest Group on Under-resourced Languages @ LREC-COLING 2024 (pp. 1-9).

---

> > ### Comment · Area_Chair_ehHp · 2025-08-07
> > **Please Engage in Author Response Discussion**
> >
> > Dear Reviewer TBJw,
> >
> > We encourage you to review the authors’ rebuttals and see how they’ve addressed your comments. If you’ve already done so, thank you! Kindly confirm your engagement by reacting in this thread or leaving a brief comment.
> >
> > Your participation helps ensure a fair and thoughtful review process.
> >
> > Best regards,
> > AC

---

### Decision · Program_Chairs · 2025-09-17

**Decision:**

Accept (poster)

**Comment:**

This paper investigates token premiums, defined as disparities in the number of tokens required to encode parallel text across different languages. High token premiums can reduce training throughput and increase inference costs.

The reviewers highly appreciated the large-scale, systematic experiments, which involved training 7k tokenizers for 98 languages. The proposed approaches for mitigating token premiums were regarded as important. One reviewer raised a concern about potential data leakage; however, the author rebuttal clarified that steps were taken to avoid leakage. The reviewer accepted this clarification and indicated they would raise their score, though the change was not reflected in the system. This has been taken into account in the meta-review.

A recurring concern among reviewers is the absence of an end-to-end evaluation to directly assess the downstream impact of reducing token premiums. While this limits the scope of the current study, the work nonetheless provides valuable insights and represents an interesting and worthwhile contribution to the NeurIPS community.